# The ALS/FTD-related *C9orf72* hexanucleotide repeat expansion forms RNA condensates through multimolecular G-quadruplexes

Federica Raguseo[1,2,3], Yiran Wang[4,5], Jessica Li[4,5], Marija Petrić Howe[4,5], Rubika Balendra[4,5], Anouk Huyghebaert [1,3], Devkee M. Vadukul[1], Diana A. Tanase [1,2], Thomas E. Maher [1,3], Layla Malouf [1,2], Roger Rubio-Sánchez [2], Francesco A. Aprile [1,3], Yuval Elani[6], Rickie Patani [4,5]✉, Lorenzo Di Michele [1,2]✉ & Marco Di Antonio [1,3,4]✉

Amyotrophic lateral sclerosis (ALS) and frontotemporal dementia (FTD) are neurodegenerative diseases that exist on a clinico-pathogenetic spectrum, designated ALS/FTD. The most common genetic cause of ALS/FTD is expansion of the intronic hexanucleotide repeat $(GGGGCC)_n$ in *C9orf72*. Here, we investigate the formation of nucleic acid secondary structures in these expansion repeats, and their role in generating condensates characteristic of ALS/FTD. We observe significant aggregation of the hexanucleotide sequence $(GGGGCC)_n$, which we associate to the formation of multimolecular G-quadruplexes (mG4s) by using a range of biophysical techniques. Exposing the condensates to G4-unfolding conditions leads to prompt disassembly, highlighting the key role of mG4-formation in the condensation process. We further validate the biological relevance of our findings by detecting an increased prevalence of G4-structures in *C9orf72* mutant human motor neurons when compared to healthy motor neurons by staining with a G4-selective fluorescent probe, revealing signal in putative condensates. Our findings strongly suggest that RNA G-rich repetitive sequences can form protein-free condensates sustained by multimolecular G-quadruplexes, highlighting their potential relevance as therapeutic targets for *C9orf72* mutation-related ALS/FTD.

Amyotrophic lateral sclerosis (ALS) is rapidly progressive, uniformly fatal and untreatable due largely to an incomplete understanding of disease mechanisms. The lifetime risk of ALS is 1:300–1:400, and over an aggressive disease course, patients become paralysed, unable to eat, speak and breathe with an average survival of between 3–5 years[1]. Frontotemporal dementia (FTD) is the second most common cause of

dementia in patients less than 65 years old and is increasingly recognised to share clinical, genetic and pathomechanistic features with ALS, termed ALS/FTD[2]. ALS and FTD, much like other neurodegenerative diseases, are characterised by the presence of pathological aggregates in neurons. Many existing studies have predominantly focused on the protein component as the leading aggregation trigger[3],

[1]Imperial College London, Department of Chemistry, Molecular Sciences Research Hub, 82 Wood Lane, London W12 0BZ, UK. [2]University of Cambridge, Department of Chemical Engineering and Biotechnology, Philippa Fawcett Drive, Cambridge CB3 0AS, UK. [3]Imperial College London, Institute of Chemical Biology, Molecular Sciences Research Hub, 82 Wood Lane, London W12 0BZ, UK. [4]The Francis Crick Institute, 1 Midland Road, London NW1 1AT, UK. [5]Department of Neuromuscular Diseases, Queen Square Institute of Neurology, University College London, London WC1N 3BG, UK. [6]Imperial College London, Department of Chemical Engineering, South Kensington, London SW7 2AZ, UK. ✉e-mail: rickie.patani@ucl.ac.uk; ld389@cam.ac.uk; m.di-antonio@imperial.ac.uk

neglecting the role of nucleic acids as a key driver in the aggregation process[4]. Indeed, it is well established in ALS/FTD that RNA-binding proteins (RBPs) such as TDP-43 and FUS possess low-complexity domains which are prone to aggregation, due to low-affinity interactions[2,5,6]. In turn, the recruitment of RNA in the aggregates has often been regarded as a secondary step of indeterminate significance. Nevertheless, the most common hereditary cause of ALS/FTD is expansion of the intronic hexanucleotide repeat (GGGGCC)$_n$ in *C9orf72*, accounting for about 40% of familial ALS cases (where the patient has a family history of ALS) and 7% of the sporadic cases (no family history of ALS) in Europe[7]. This mutation has been previously proposed to lead to the formation of aggregates *via* various mechanisms[8,9], including RNA transcribed from the expansion repeat playing a structural role. However, a clear mechanism to support this hypothesis has yet to be described[10,11].

The hypothesis that RNA expansion repeats can aggregate in the absence of additional proteins has gained significant traction in recent years[12–15]. Although endogenous nucleic acid sequences do not typically lead to aggregation, they have the potential to form multimeric networks through both canonical and non-canonical base pairing interactions[16]. For example, r(CAG)$_n$ and r(GGGGCC)$_n$, both natural sequences related to neurodegenerative diseases (spinocerebellar ataxia[17] and ALS/FTD respectively), phase-separate in vitro at a critical number of repeats, suggesting that they can form multimolecular structures[18].

r(GGGGCC)$_n$, in particular, has previously been shown to arrange into hairpin and G-quadruplex (G4) structures, both of which have exhibited potential involvement in disease progression[19]. Indeed, the

potential of these structures as therapeutic targets is demonstrated by using small molecule probes that bind to and stabilise them, which leads to amelioration in disease models[11,19–23]. In particular, the use of ligands to bind G4s has been shown to ameliorate ALS phenotypes in neuronal cells[10,19,20,22] and targeting of the hairpin with a small molecule inhibited repeat-associated non-ATG (RAN) translation and subsequent generation of toxic dipeptide repeats from the *C9orf72* gene mutation[11]. However, none of these studies have fully clarified the role of the nucleic acid structures in the regulation of pathological aggregation, which is a pre-requisite to devising optimised therapeutic agents. Furthermore, controversy exists about which secondary structure, G-quadruplex or hairpin, is most relevant to disease progression, and how. In this study, we attempted to address this controversy by investigating the role of an alternative structure that can be formed by the (GGGGCC)$_n$ repeat, which has not been studied in the ALS/FTD context: a multimolecular G-quadruplex structure (mG4). We hypothesised that multimeric G-quadruplexes could provide three-dimensional linkages by the formation of G-G base pairing, which could efficiently initiate biomolecular condensates in *C9orf72* mutation-related ALS/ FTD.

Indeed, non-canonical G-G base pairing underpinning G4-formation can readily occur under physiological conditions[24–26] (Fig. 1a) and G4-formation has already been associated with cancer, neurodegenerative disease (ALS/FTD) and other rare-genetic diseases (Fragile X syndrome and Cockayne Syndrome)[27,28]. The hexanucleotide intronic repeat (GGGGCC)$_n$ in the *C9orf72* gene has been shown to form G4s both in its RNA and DNA form[29–32] and the RNA repeat r(GGGGCC)$_n$ has been visualised within cellular ALS aggregates[20]. In

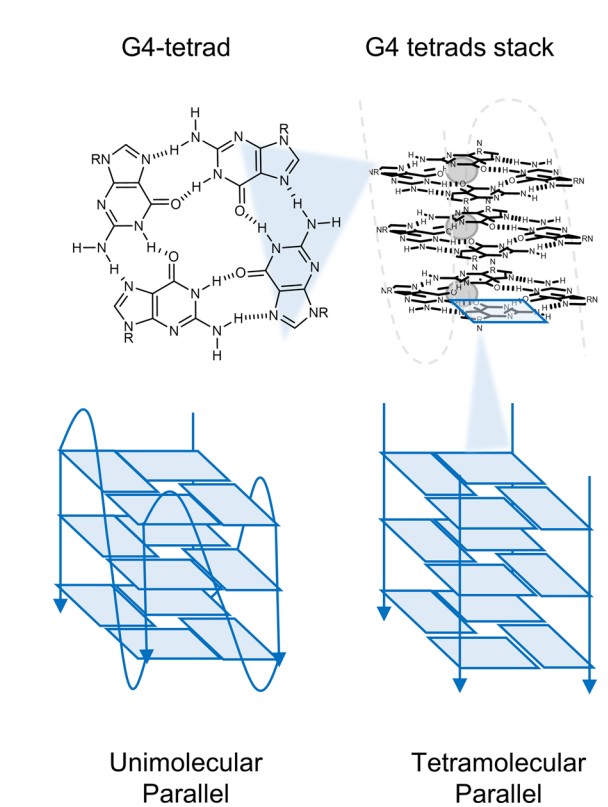

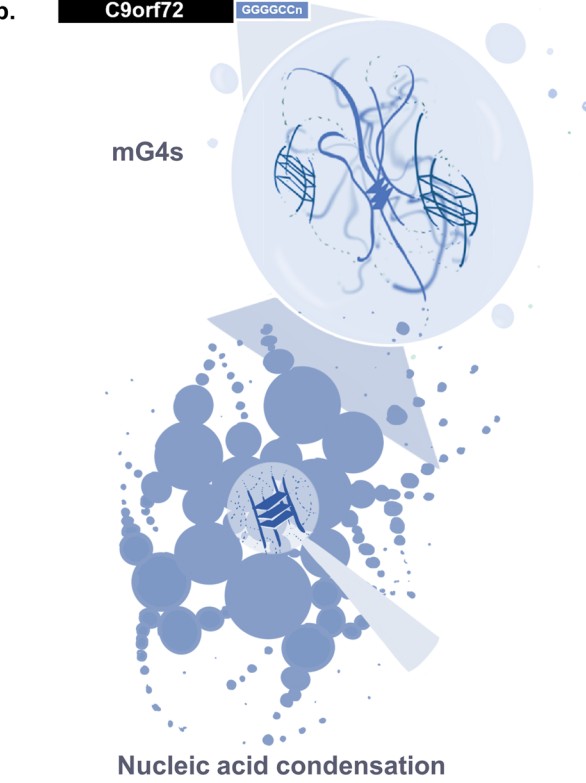

**Fig. 1 | G-quadruplexes structural features and mechanistic hypothesis.**
**a** Structural characteristics of G-quadruplexes—The G4 scaffold is composed of a G-tetrad, a planar tetrameric assembly of guanine (G) bases held together by Hoogsteen H-bonding. G4-tetrads stack to form a typical G4-structure, which is stabilised by monovalent cations that can coordinate the G-tetrad, which is dependent on the size of the cation (Li$^+$ » Na$^+$ > K$^+$). G-tetrads can assemble into

G4-structures either within the same DNA/RNA strand (unimolecular) or by condensation of multiple independent DNA/RNA strands (multimolecular, *i.e.*, tetramolecular from four strands). **b** Proposed mechanism of aggregation in ALS/FTD pathological aggregates—The (GGGGCC)$_n$ repeat sequence can aggregate in the absence of proteins due to the formation of multimolecular G4s (mG4s).

addition, TDP-43 and FUS proteins, two RNA-binding proteins present in pathological aggregates in ALS (TDP-43 in >97% of cases[33]) and FTD (TDP-43 in approximately 40% of cases[34] and FUS in approximately 10% of cases[35]), have both shown G4-binding abilities, reinforcing the idea that G4s might play a key role in the formation of condensates typical of ALS/FTD[36–38]. Furthermore, the hypothesis that mG4s are key to pathological aggregation would imply that targeting the hairpin conformation would slow disease progression by locking the RNA repeat in the hairpin conformation, preventing aggregate formation. Similarly, treatment with G4-ligands could promote formation of unimolecular G4s, also preventing formation of mG4s and thus reducing the size of condensates upon treatment with G4-ligands, which would reconcile the seemingly discordant observations in the literature.

To investigate this, we extensively characterised the biophysical behaviour of $(GGGGCC)_n$ repeats to determine their ability to form mG4s and mG4-condensates under physiological conditions, observing substantial aggregation dependent on both repeat length $n$ and oligonucleotide concentration. Next, we demonstrated that mG4s are structurally essential to condensation, as exposure to G4-unfolding conditions led to prompt condensate disassembly. Consistently, we observed that condensation is hindered by G4-ligand PDS, which has been previously suggested to favour binding to unimolecular over multimolecular G4s[39], and by rational design of mutants of the $(GGGGCC)_n$ repeat. Our results suggest that the *C9orf72* transcript itself aggregates through mG4s formation (Fig. 1b). Furthermore, we have demonstrated that mG4 mediated condensation can be enhanced by the presence of the ALS/FTD related protein TDP-43, further indicating the potential therapeutic relevance of our findings. Finally, we used the G4-selective fluorescent probe NMM to detect G4-structures in *C9orf72* mutant ALS/FTD-patient derived motor neurons, revealing a statistically significant increase in G4-prevalence within *C9orf72* mutant motor neurons when compared to healthy counterparts. Cumulatively, our data provide novel mechanistic insight into pathogenic biomolecular condensation, a potential precursor to insoluble aggregate formation in ALS/FTD. Our findings may help to inform strategy for therapeutic targeting in these currently untreatable diseases.

## Results

To assess if multimolecular G4s (mG4s) can promote condensation of the hexanucleotide repeat sequence $(GGGGCC)_n$, we started by investigating the ability of $(GGGGCC)_n$ to form mG4s under physiological conditions and their stability to generate strong cross-links between nucleic acids strands.

### $(GGGGCC)_n$ forms mG4s

We first investigated the ability of the $(GGGGCC)_n$ repeats to form multimolecular G4-structures. Although RNA is recognised as one of the toxic species in *C9orf72* mutation-related ALS/FTD pathological aggregates, we performed our initial biophysical studies with the DNA equivalents, as they are more easily sourced and handled[40]. To promote G4-formation, we annealed DNA $(GGGGCC)_n$ repeats of variable lengths ($n = 1$ to 12) in a $K^+$-containing buffer (500 mM KCl). To mimic intra-cellular crowding, 30% polyethylene glycol (PEG) was included in the buffer prior to slowly annealing the samples from 95 °C to 25 °C at a rate of $-0.02$ °C/min (see methods for details).

Circular dichroism (CD) spectroscopy was initially performed to assess the formation of G4-structures. CD spectra of the $(GGGGCC)_n$ repeats annealed under mG4-forming conditions presented the characteristic negative peak at ~240 nm and the positive peak at ~263 nm, which are typical of a parallel G4-topology (Fig. 2a)[41]. Although CD confirmed G4-formation, it cannot distinguish unimolecular from multimolecular topologies. Agarose gel electrophoresis (AGE) was thus employed to assess mG4-formation (Fig. 2b). Two main bands are observed, which can be ascribed to the unimolecular and bimolecular

species based on previous literature[28,31] (Fig. 2b). It is noteworthy that the gel also indicated formation of bands at higher molecularity, which could not be visualised due to the low SYBR safe staining efficiency. To address this, we demonstrated formation of multimeric species that could be observed by AGE using carboxyfluorescein (FAM)- labelled DNA (FAM) at lower concentrations to avoid smearing, as shown in Fig. S1.

The combination of CD and gel electrophoresis results suggested the ability of $(GGGGCC)_n$ sequences to fold into parallel G4s that can also adopt a multimolecular stoichiometry. To further demonstrate that the multimolecular species observed in the agarose gel could be ascribed to mG4s, we performed a KCl titration (0–500 mM) experiment, as multimolecular G4-formation is highly dependent on the concentration of $K^+$. AGE experiments revealed that increasing $K^+$ concentrations can quantitatively convert the fast-running band observed in Fig. 2b, which we ascribed to the unimolecular G4, to a slowly running dimeric band, which is consistent with the formation of a bimolecular G4-structures (Fig. 2c).

To further verify that the slow running bands observed by AGE contained mG4s, we also stained the samples with N-methyl meso-porphyrin IX (NMM), a G4-specific fluorescent dye[42–44]. NMM is expected to fluoresce upon binding to the bands containing G4s but not the non-G4 DNA control. Indeed, NMM selectively stains the slowly migrating portions of the $(GGGGCC)_n$ bands, indicating that the higher molecular weight bands contain G4s, as displayed in Fig. 2d. To corroborate this observation, the AGE experiment was repeated using NMM staining, which confirmed the presence of G4s in the slow-moving band. The intensity of NMM fluorescence recorded in the slow-moving bands increases with $K^+$, consistent with a greater abundance of (multimolecular) G4 species (Fig. 2c).

### mG4s lead to DNA condensation in absence of proteins

Having demonstrated the formation of multimolecular G4s in DNA samples of $(GGGGCC)_n$, that increase in size with $K^+$ concentration and length of the repeat, we proceeded to systematically screen for the emergence of macroscopic condensates formed by these repeats using confocal microscopy (see methods). Figure 3a shows representative images of aggregates that are generated for different repeat lengths at a fixed concentration of 250 μM and that can be easily detected by confocal microscopy. Consistent with the observed clinical correlation between repeat length and pathological condensation in ALS/FTD[7], we observed that in vitro condensation is more prominent for longer repeats. Furthermore, the condensates generated by a FAM-fluorescently labelled $(GGGGCC)_{10}$ repeat showed no fluorescence recovery after photobleaching, indicating a solid or gel-like state (Fig. S2C).

The phase diagram in Fig. 3d and Fig. S3 systematically maps the condensation state of the system as a function of repeat length and oligonucleotide concentration. Strands with repeat lengths of 2 and 3 showed no condensation even at the highest tested concentration of 500 μM. The shortest repeat-length producing condensates at 500 μM is $n = 4$, while constructs with $n \geq 9$ aggregated at oligonucleotide concentrations as low as 50 μM. The strong dependence of the phase-boundary location on repeat length confirms the positive correlation that an increased repeat length has on the formation of multimeric structures, which is also consistent with AGE data in Fig. 2b, d.

Microscopy results thus confirm that $(GGGGCC)_n$ can form condensates in the absence of proteins. Staining condensates formed by $(GGGGCC)_6$ with NMM, expectedly produced a detectable fluorescence signal, indicating the presence of G4s in the condensates (Fig. 3c). The selectivity of NMM fluorescence for G4-based condensates when compared to canonical dsDNA was next assessed (Fig. S4) under the same conditions. As expected, no fluorescence was detected in DNA condensates that lack G4s (Fig. S4B).

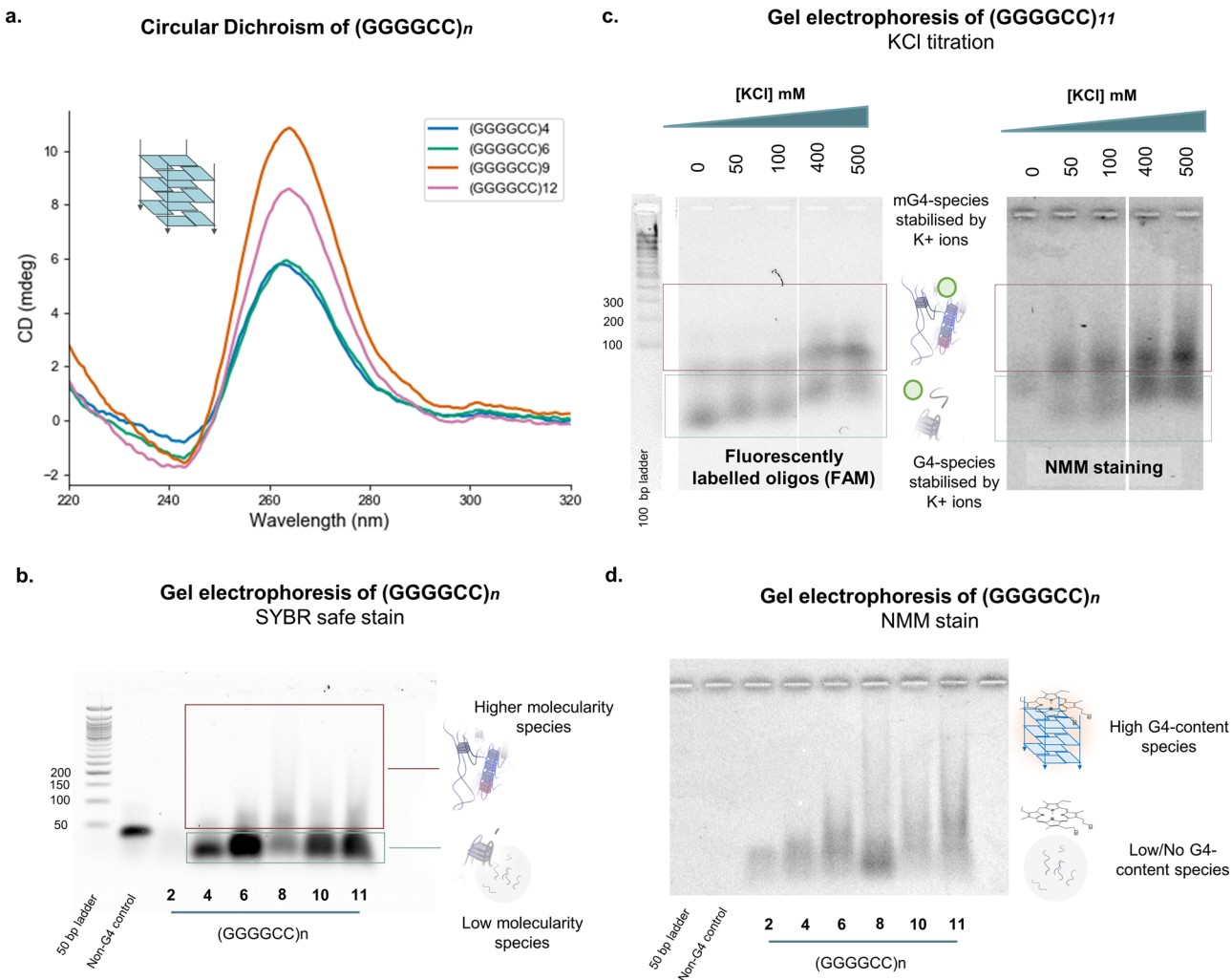

**Fig. 2 | DNA (GGGGCC)$_n$ forms mG4s at different repeat lengths. a** (GGGGCC)$_n$ CD spectra−(GGGGCC)$_n$ ($n$ = 4, 6, 9, 12) were annealed at 500 μM, 250 μM, 100 μM and 50 μM under mG4-forming conditions (see methods) and further diluted to 5 μM prior CD analysis. All the samples present a positive peak at about 263 nm and a negative peak at 240 nm that can be associated to a parallel G4 conformation. **b** (GGGGCC)$_n$ agarose gel electrophoresis−SYBR safe stain. (GGGGCC)$_n$ ($n$ = 2-11) were annealed at 250 μM under mG4-forming conditions (see methods) and the gel was stained with SYBR safe. This gel includes the same samples used for the following NMM staining displayed in (**d**) and has been added for reference to aid in the visualisation of the location of the bands. **c** (GGGGCC)$_{11}$ agarose gel electrophoresis −(GGGGCC)$_{11}$ was annealed in the presence of different KCl concentrations. As KCl concentration increases, so does the molecularity of the species involved, indicating that the formed multimolecular structures have a strong KCl dependence. In the schematics, the green ball represents the K+ ion. On the right, the same samples are stained with NMM, while on the left the sequences are FAM labelled. For reference a 100 base pairs ladder has also been added, which was stained with SYBR safe. **d** (GGGGCC)$_n$ agarose gel electrophoresis−NMM stain. (GGGGCC)$_n$ ($n$ = 2-11) were annealed at 250 μM in mG4-forming conditions. The lanes containing any of the (GGGGCC)$_n$ were fluorescent upon staining with NMM, implying the presence of G4s in the higher molecularity species formed under these conditions. The absence of staining for both the ladder and the non-G4 ssDNA control confirm the specificity of the dye for G4-containing species.

We, therefore, hypothesised that macroscopic aggregation was mediated by formation of mG4s causing the condensation of oligonucleotides. However, NMM staining does not per se imply a structural role of quadruplexes but only demonstrates their presence within the condensate. To assess whether G4s play a structural role in the formation of the observed macroscopic condensates, we performed NMM photo-oxidation experiments (see methods). NMM binds to G4s by end-stacking and, upon irradiation with a high-power laser, it triggers guanine photo-oxidation, leading to the formation of 8-oxoguanine that causes the disassembly of the G4-structure[45]. This phenomenon has been previously observed to not occur in dsDNA[45], making it a highly G4-specific disassembly route, which we have previously leveraged to achieve light-controlled disassembly of mG4s in engineered DNA nanostructures[46]. As shown in Fig. 3e, we irradiated a portion of the (GGGGCC)$_6$ NMM-stained condensates to trigger photo-oxidation, which led to immediate disassembly of the condensates.

Given that no other relevant multimolecular interaction is known to be affected by NMM photo-oxidation, these results suggest that mG4s are the prevalent structure holding the condensate together and that in the absence of guanine-guanine hydrogen bonding, necessary for G4-formation, no aggregation is observed. Moreover, we tested the selectivity of the assay on NMM-stained dsDNA condensates, which did not show any significant change upon irradiation (Fig. S4C). Notably, although the samples analysed were annealed in 500 mM KCl to promote mG4-formation in high yields, condensates were also observed at physiological KCl amounts at higher DNA concentrations (Fig. S5). Additionally, to ensure that the crowding agent was not triggering DNA condensation in its own right, (GGGGCC)$_{11}$ annealing was also performed in absence of PEG with a slower cooling rate to promote mG4-formation. As shown in Fig. S6, condensates were still detected in the absence of PEG, confirming that the crowding agent was not influencing DNA condensation.

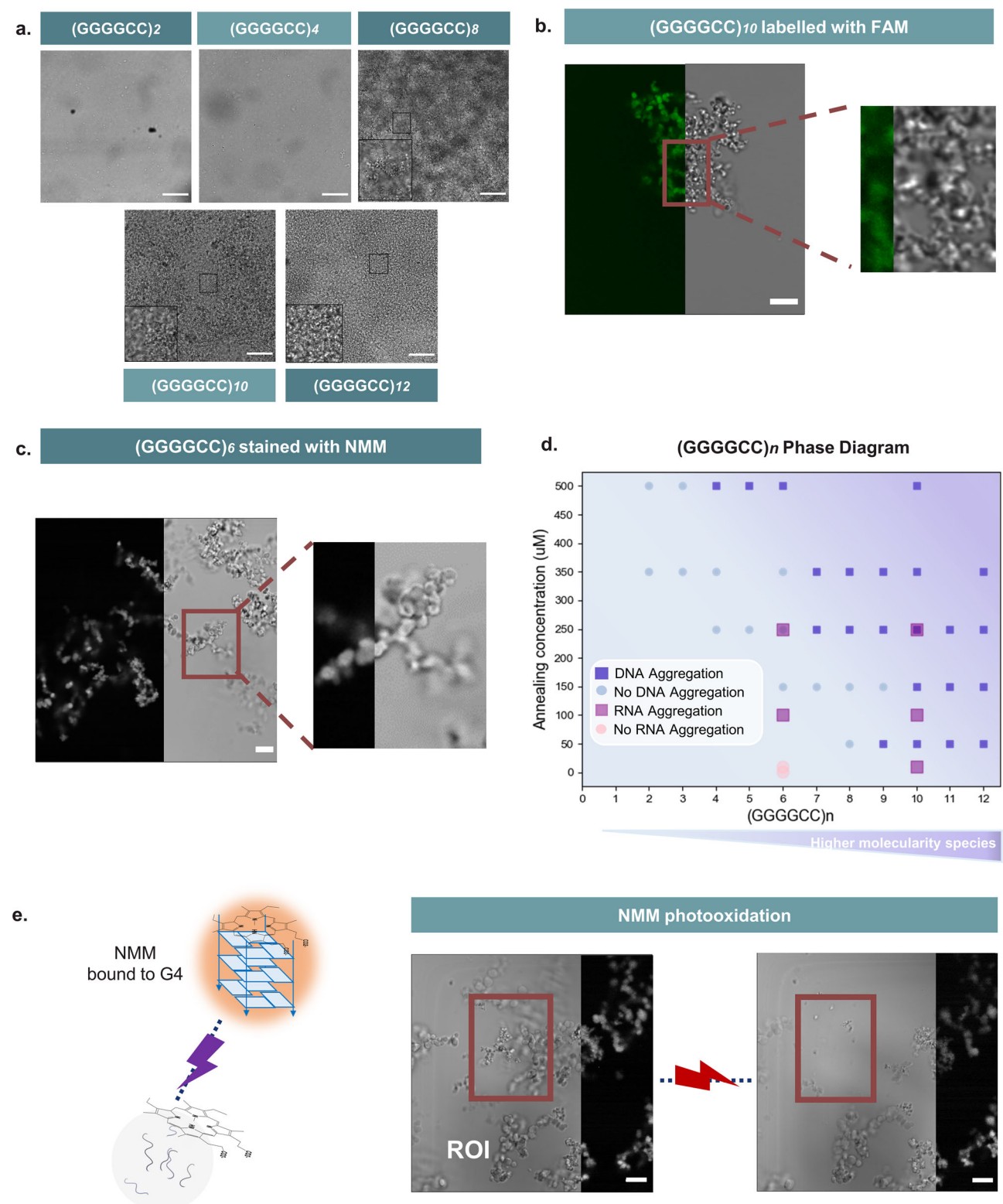

## (GGGGCC)$_n$ mutants that prevent mG4-formation show reduced aggregation

To further confirm that mG4s were responsible for condensation observed for (GGGGCC)$_n$ repeats, we investigated the condensation properties of a series of mutated sequences (Table 1).

In addition to G4-formation, it is possible that GC base pairing may play a role in stabilising the observed aggregates. To test this hypothesis, we studied the (CCCCGG)$_6$ repeat, which retains the same number of potential canonical base pairing interactions as (GGGGCC)$_n$ but is unable to form G4s. CD was initially performed to confirm the absence of G4-formation, which was further validated via NMM staining of these sequences loaded on an agarose gel (Fig. 4a, b, c). Replacing Cs with As in the (GGGGAA)$_6$ repeat preserves G4-forming ability but prevents the formation of canonical base pairing. CD of this repeat revealed a typical parallel-G4 trace and NMM staining of the agarose gel confirmed G4 formation (Fig. 4a, b). Similarly, (GGGGTT)$_6$

**Fig. 3 | DNA (GGGGCC)$_n$ aggregates in a G4-dependent fashion. a** (GGGGCC)$_n$ aggregates—Brightfield imaging of (GGGGCC)$_n$ ($n$ = 2–12) annealed under mG4-forming conditions at 250 μM (see methods). 100 μm scalebar. Micrographs in (**a**) are representative from $n$ = 2 independent replicates. **b** FAM labelled (GGGGCC)$_{10}$—FAM-labelled (GGGGCC)$_{10}$ was annealed under mG4-forming conditions (see methods). The image is split to show the brightfield channel (on the right) and the FAM (on the left). 10 μm scalebar. Data for (**b**) is a representative micrograph from $n$ = 2 independent replicates. **c** (GGGGCC)$_6$ NMM staining experiment—(GGGGCC)$_6$ was annealed under mG4-forming conditions and stained with NMM (see methods). The image is split to show the brightfield channel (on the right) and the NMM fluorescence channel (on the left). Data from (**c**) is a representative micrograph from $n$ = 2 independent replicates. 10 μm scalebar. **d** (GGGGCC)$_n$ phase diagram in KCl—(GGGGCC)$_n$ was annealed under mG4-forming conditions. At lower annealing concentrations and lower repeat lengths, the sequence does not present the ability to aggregate. The higher the repeat length and the higher the concentration, the more likely it is to observe aggregation. r(GGGGCC)$_{6,10}$ were also annealed under mG4s forming conditions and presented aggregation at lower concentrations than their DNA equivalents. A colour gradient is utilised in the phase diagram to symbolise a higher likelihood of finding aggregates for a given sequence. **e** (GGGGCC)$_6$ NMM photooxidation experiment—Upon laser excitation the NMM dye bound to the mG4s photo-oxidises the guanine in the quadruplex, leading to the disassembly of the structure. The region of interest highlighted in the figure (ROI) was irradiated for 1 min and the second image was subsequently acquired, revealing disassembly of the aggregates within the ROI due to photooxidation. The image is split to show the brightfield channel (on the left) and the NMM (on the right). 10 μm scalebar. Data from (**e**) is a representative micrograph from $n$ = 2 independent replicates.

maintains the ability to form G4s preventing base pairing, but due to G-T stacking interactions it is expected to favour mono-molecular G4-structures over multimolecular ones[47], allowing us to disentangle the effect of unimolecular vs multimolecular G4s in the aggregation process. CD and agarose gel-NMM staining of the T-to-C mutant indeed revealed formation of G4s (Fig. 4b, c).

Having gauged the G4-forming tendencies of the mutants, we proceeded to assess their condensation properties. No significant condensation was observed for (CCCCGG)$_6$, confirming that canonical base pairing interactions and hairpin formation are not sufficient to drive condensation of the oligonucleotides. Similarly, no significant condensation was observed for (GGGGTT)$_6$, which predominantly forms unimolecular G4s as opposed to mG4s (see AGE in Fig. 4b). This evidence further supports our hypothesis that mG4s are the essential cross-linking elements in the condensates, while formation of G4s per se is insufficient to prompt condensation. Consistently, (GGGGAA)$_6$ and (GGGGCC)$_6$ were the only two sequences to reveal significant mG4-formation in the AGE experiments, and to show formation of condensates under confocal microscopy (Fig. 4d).

### Pyridostatin prevents (GGGGCC)$_n$ aggregation

G4 ligands have been shown to ameliorate the presence of ALS aggregates in cells[20,48]. The ligands have been speculated to act by disrupting the toxic downstream functions of *C9orf72* thought to underpin aggregation, such as preventing binding to RBP[48]. Having demonstrated that mG4s alone are able to prompt aggregation in protein-free samples, we asked whether G4 ligands could prevent condensation by promoting the formation of unimolecular G4s as opposed to multimolecular ones, rather than displacing RBP interactions. For this purpose, we employed pyridostatin (PDS), a well-characterised G4-ligand[39], known to form more stable complexes with lower molecularity G4s over multimolecular G4s[49] and compatible with our experimental conditions.

NMM-stained AGE of (GGGGCC)$_{10}$ annealed in the presence of 10 μM PDS revealed a substantial reduction in high molecularity bands compared to the PDS-free sample (Fig. S7). The ability of PDS to suppress large multimers is reflected by a substantial reduction in the number and size of condensates visible by confocal microscopy (Fig. 5; see methods). It is worth noting that the small percentage of DMSO introduced to solubilise the ligand does not directly impact aggregation, as displayed in Fig. 5.

Beyond further confirming the critical importance of G4 molecularity in stabilising (GGGGCC)$_n$ aggregates, these data might underpin a potential new mechanism for the reported ability of G4 ligands to ameliorate reported ALS phenotypes in cellular models[20].

### RNA repeats form biomolecular condensates at lower concentrations than their DNA counterparts

RNA is considered a candidate toxic species in *C9orf72* mutation-related ALS and FTD pathological aggregates. As DNA equivalents were tested in the preceding experiments, we sought to address whether RNA repeats conformed to similar behaviour and whether any notable differences existed. Indeed, RNA G4s (rG4s) are known to arrange preferentially into a parallel conformation and to be more thermodynamically stable than the equivalent DNA G4s[50,51]. For these reasons, we hypothesised that aggregation of r(GGGGCC)$_n$ could occur at lower concentrations in RNA compared to with DNA but that, overall, the aggregates would possess similar physical and chemical properties.

We sought to confirm the ability of RNA to form condensates in a similar manner to its DNA equivalent by subjecting r(GGGGCC)$_{6/10}$ to the same condensation conditions. Aggregation was observed for concentrations as low as 100 μM for repeat length 6, and 10 μM for $n$ = 10 (Fig. S8), forming amorphous aggregates similar in morphology to their DNA equivalents under all conditions. For comparison, DNA (GGGGCC)$_6$ was seen to aggregate only above 350 μM in concentration, reflecting the known ability of RNA to form more stable G4s compared to DNA. To further validate mG4-formation in the RNA aggregates, samples were stained with NMM, confirming the presence of G4s (Fig. 6a). Furthermore, the RNA aggregates stained with NMM were subjected to irradiation at 405 nm to induce guanine photo-oxidation, which caused immediate disassembly of the irradiated aggregates (Fig. 6b), as previously observed for DNA. These results confirmed that (GGGGCC)$_n$ aggregation is an mG4-dependant phenomenon and occurs in a similar manner for RNA and DNA repeats, but crucially occurs at lower concentrations in the case of RNA.

### (GGGGCC)$_n$ enhances aggregation of TDP-43

Transactive response DNA binding protein of 43 kDa (TDP-43) is a nucleic acid-binding protein that plays a crucial role in RNA processing and regulation of gene expression and has been extensively studied in the context of neurodegenerative diseases, particularly in ALS/FTD where it forms pathological aggregates in affected neurons[52]. Interestingly, this protein has previously shown G4-binding abilities, raising the possibility that its interaction with G4s enhances or even triggers pathological aggregation[36].

To test this hypothesis, we purified a TDP-43 protein variant which only comprises the RNA binding region, *i.e.*, RRM1–2 (K102–Q269)[53] (Fig. S9). We selected this variant because it binds to nucleic acids similarly to full-length TDP-43 but is substantially easier to express and purify. Furthermore, it is more soluble than full length TDP-43 in vitro after purification, yet it can form aggregates under specific conditions that resemble physiology[53]. The aggregation propensity of this variant

### Table 1 | Table of mutants

| DNA sequence | G4s | Canonical base pairing | Aggregation |
|---|---|---|---|
| (GGGGCC)$_6$ | Yes | Yes | Yes |
| (GGGG**TT**)$_6$ | Yes | No | No |
| (GGGG**AA**)$_6$ | Yes | No | Yes |
| (**CCCCGG**)$_6$ | No | Yes | No |

The mutated bases are highlighted in bold

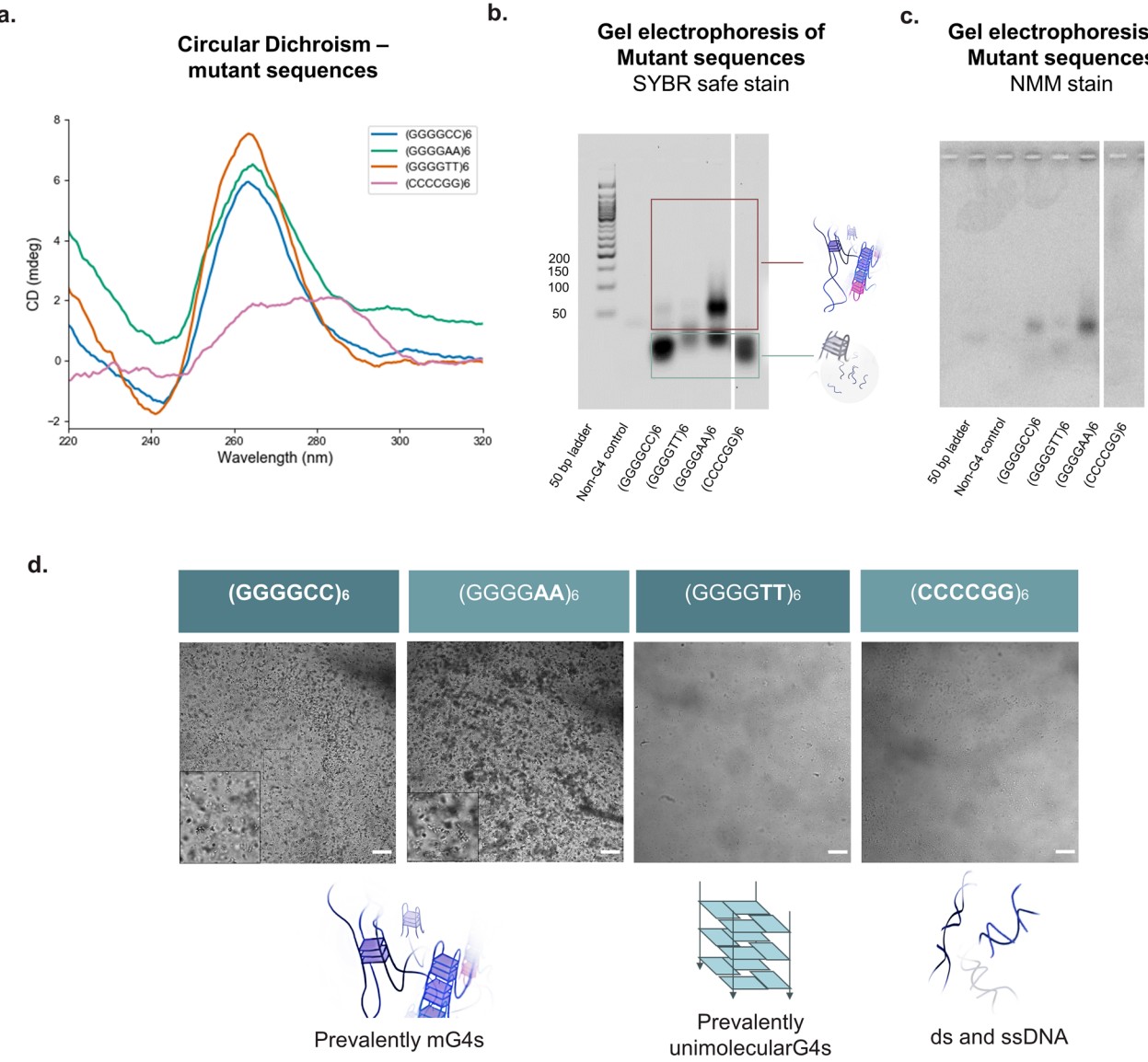

**Fig. 4 | Role of base pairing and monomolecular G4s in the aggregation mechanism. a** Circular Dichroism spectra of mutants−Controls of $n = 6$ were annealed under mG4-forming conditions (see methods). Sequences forming G4-structures present the characteristic peaks at ~263 nm and 240 nm, whilst the $(CCCCGG)_n$ control fails to generate any significant CD signal. **b** Mutants agarose gel electrophoresis−SYBR safe stain−Controls of $n = 6$ were annealed under mG4-forming conditions. **c** $(GGGGCC)_n$ agarose gel electrophoresis−NMM stain−Controls of $n = 6$ were annealed under mG4-forming conditions. All G4-forming sequences $((GGGGAA)_6, (GGGGTT)_6, (GGGGCC)_6)$ presented fluorescence bands upon NMM staining, confirming the presence of G4-structures as observed by CD. Ladder, non-G4 ssDNA control, and $(CCCCGG)_6$ do not yield any NMM staining and are not visible in the gel, confirming the specificity of the dye for G4-containing species. **d** Aggregation of the mutant sequences−Controls of $n = 6$ were annealed under mG4-forming conditions. Aggregates were only detected under confocal microscopy for the $(GGGGAA)_6$ and $(GGGGCC)_6$ sequences, which are the only ones able to form mG4s. 100 μm scalebar.

was sufficient to accurately permit the assessment of protein and mG4 condensation over a range of concentrations. First, we assessed whether the protein could aggregate in the presence of the $(GGGGCC)_n$ repeat. To achieve this, we annealed FAM-labelled $(GGGGCC)_{11}$ under mG4-forming conditions (100 μM, see methods) and incubated it for 3 days with 10 μM of TDP-43 RRM1−2 labelled with ALEXA 633. The resulting aggregates formed were active in both fluorescence channels (Fig. S10A), suggesting that TDP-43 RRM1−2 was co-aggregating with the $(GGGGCC)_n$ repeat. We then investigated how the presence of TDP-43 RRM1−2 impacts $(GGGGCC)_{11}$ aggregation. To study this, we screened the aggregation properties for a series of $(GGGGCC)_{11}$ concentrations (0, 1, 100, 300 μM) in the presence or in the absence of a fixed amount of TDP-43 RRM1−2 (10 μM).

As shown in Fig. 6c, we detected more condensation when the $(GGGGCC)_{11}$ repeat was exposed to TDP-43 RRM1−2, suggesting that

the protein domain can enhance the aggregation of the oligonucleotide. Importantly, no macroscopic aggregates can be detected with TDP-43 RRM1−2 alone. This suggests that the emergence of larger aggregates requires both the protein and the nucleic acid components. The formation of TDP-43 RRM1−2 aggregates that cannot be detected by confocal microscopy was assessed via transmission electron microscopy, revealing the formation of microscopic aggregates that cannot be detected by confocal microscopy, consistently with previous literature[54] (Fig. S11). To confirm that mG4s were still present in the final protein/nucleic acids condensates, the samples were stained with the G4-specific fluorescent probe NMM. As shown by NMM staining, condensates formed in the presence of TDP-43 RRM1−2 still had a significant G4-content (Fig. S10B). Given the pathological relevance of TDP-43 aggregation in ALS/FTD, we anticipate that the observed synergy between RRM1−2 and $(GGGGCC)_n$ repeats bear

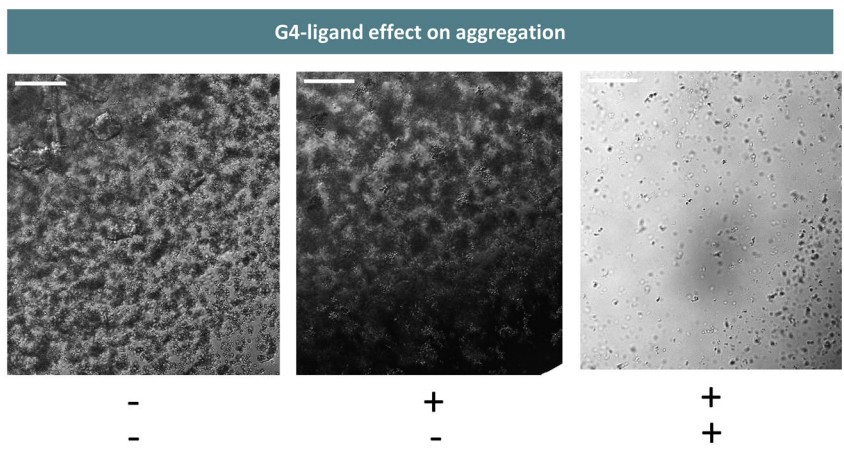

**Fig. 5 | (GGGGCC)$_{10}$ aggregation in presence of 10 μM PDS.** (GGGGCC)$_{10}$ was annealed at 150 μM under mG4-forming conditions. The sample containing the G4-ligand presents less aggregates and of a smaller size with respect to the non-PDS containing controls. 100 μm scalebar. The micrographs are representative micrographs of $n = 2$ independent experiments.

significance for the aggregation observed in ALS/FTD and may inform the development of future therapeutic strategies.

## G4-aggregates can be visualised by NMM in *C9orf72* mutant patient derived neurons

Encouraged by the enhanced TDP-43 aggregation observed in the presence of the (GGGGCC)$_n$ repeats, we sought to further validate the physiological relevance of our findings by using ALS/FTD patient derived spinal cord motor neurons generated from induced pluripotent stem cells (iPSCs) carrying the *C9orf72* mutation. Here we used a fully established human stem cell model[55], which has previously been reported to reveal RNA *foci* that can be visualised by microscopy[20,56].

We sought to explore the prevalence of G4-structures in *C9orf72* mutant motor neurons when compared to healthy counterparts. To achieve this, we stained fixed human *C9orf72* mutant neurons and healthy motor neurons with NMM (10 μM) to visualise G4s and observe any aggregations, and the nuclear dye DRAQ5 to identify the nuclei of the cells. To confirm the purity of neuronal populations we stained the cells with β-tubulin III (Fig. S12A), which revealed highly enriched motor neuron populations in both control and mutant cultures. Additionally, DMSO-treated cells employed as a negative control showed no fluorescence in the NMM emission spectrum, indicating minimal cell autofluorescence and high signal specificity in NMM-treated cells, further corroborated by flow cytometry (Figs. S12B and S13). Treatment of *C9orf72* mutant and healthy control motor neurons with NMM revealed an efficient G4-structure labelling (Fig. 7a). A statistically significant increase in the number of NMM foci was observed both in the cell soma and within the nucleus in *C9orf72* mutant motor neurons when compared to control motor neurons (Fig. 7b, c), but not in their intensity nor size (Fig. S12E, F). Notably, no difference in total NMM signal intensity or prevalence of cells bearing NMM *foci* was detected (Fig. S12C, D). This was further corroborated by flow cytometry where control and *C9orf72* mutant motor neurons exhibited comparable NMM signal (Fig. S13). Altogether, the differences observed are consistent with a higher G4-prevalence in the *C9orf72* mutant motor neurons.

## Discussion

Although unimolecular G4s are increasingly recognised drug targets for many diseases ranging from cancer to Cockayne Syndrome[24], the biological relevance of mG4s has only recently started to be investigated[57]. Indeed, there is growing evidence of how mG4-formation could aid long-range interactions at enhancer-promoter regions in the genome[58] and how mG4s could promote formation of

phase-separated biomolecular condensates and aggregates[9,57]. Furthermore, the recent discovery of an endogenous human protein that selectively binds mG4s over unimolecular structures strongly suggests that these multimeric species could be formed under physiological conditions and play important roles in regulating cellular homoeostasis[28].

Although unimolecular G4s have a kinetic advantage and therefore form more readily, mG4s are usually thermodynamically favoured under the crowded conditions that are present within cells[59]. Despite the increasingly recognised biological role of mG4s, there is no specific tool to test for their presence in vitro or in vivo. Much like when unimolecular G4s were first discovered and there were no specific probes that would distinguish them from dsDNA in cells, none of the currently used G4-specific probes (such as PDS[60], PhenDC3[61], BG4[62] and NMM[44]) have been thoroughly studied in the context of mG4s, making their differentiation from unimolecular G4s in live-cell experiments challenging.

In this study we showed how (GGGGCC)$_n$ can arrange into mG4s *via* a combination of biophysical tools: CD demonstrating the formation of G4-structures, AGE confirming the formation of multimolecular species, and both NMM staining and KCl titrations providing further evidence to support the formation of mG4 species in *C9orf72* expansion repeats characteristic of ALS/FTD. We further observed a clear correlation between mG4-formation and the ability of (GGGGCC)$_n$ to form biomolecular condensates in vitro. Since the pathological aggregate formation in ALS/FTD is strongly correlated with the presence of (GGGGCC)$_n$ at high repeat numbers, we investigated the relationship between the observed nucleic acid-driven aggregation and repeat length itself. We were able to show that higher repeat lengths require lower concentrations for aggregation to occur, which may bear significance for the disease setting. The repeat lengths used for this study are significantly below the recognised pathological threshold ($n = 24$)[7], which is possibly why we only observe aggregation in the micromolar range. However, the phase diagram generated in this study implies that higher repeat lengths would aggregate at lower concentrations, reaching biologically relevant ranges (a human cell contains ~ 10–30 pg total RNA and 6 pg of DNA[63]).

The aggregates generated by the (GGGGCC)$_n$ repeats were amorphous and solid in appearance. Although pathological condensation in ALS and FTD are often referred to as liquid-like[64–66], it is reasonable that additional protein components and RNAs[67] in ALS/FTD-affected cells could modify the physical state of the aggregates. In fact, two of the main proteins implicated in ALS/FTD (TDP-43 and FUS) have shown significant G4-binding abilities[36,68], suggesting that

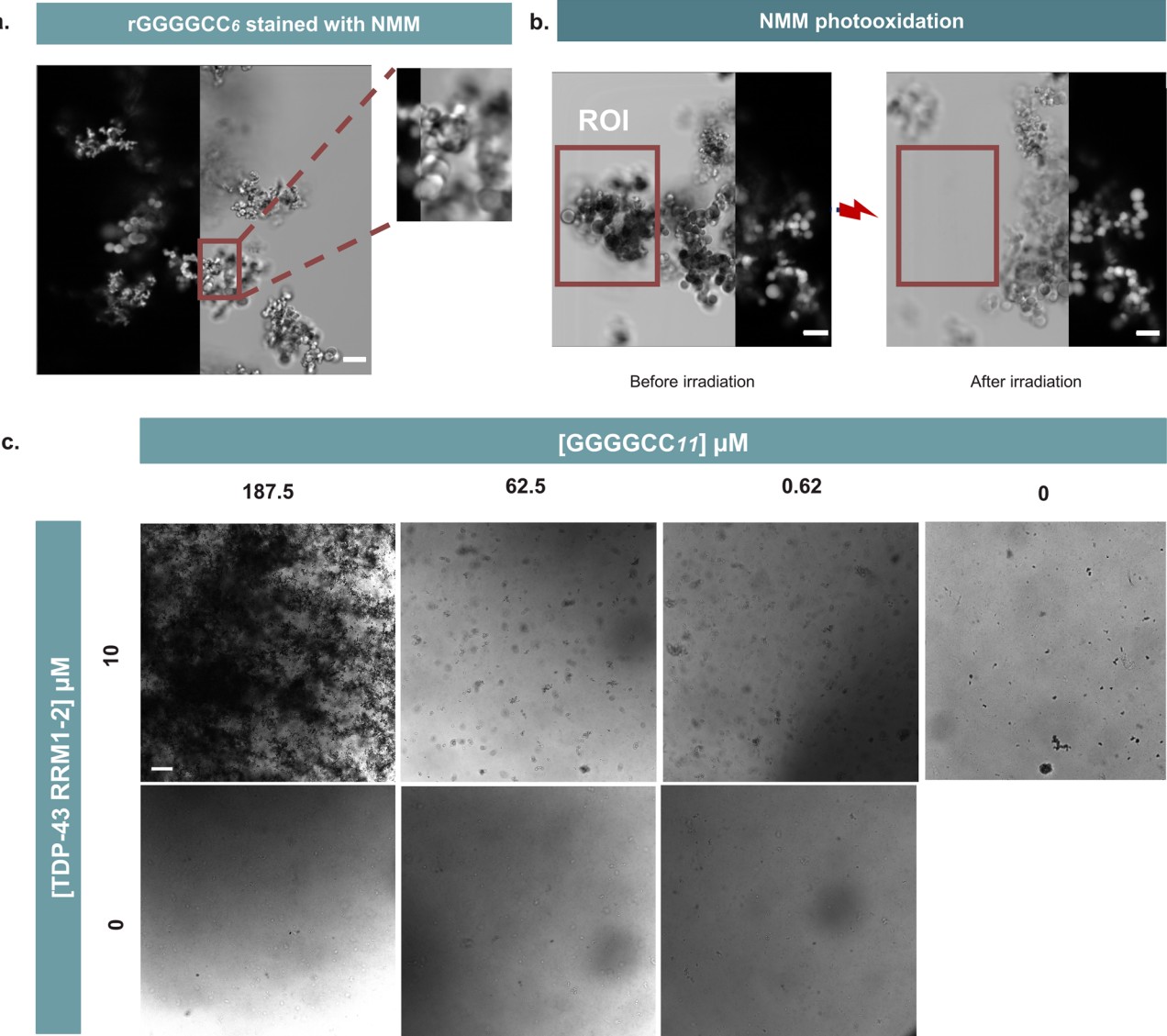

**Fig. 6 | (GGGGCC)$_n$ ability to aggregate in its RNA form and to interact with TDP-43 RRM1-2. a** RNA (GGGGCC)$_n$ aggregates in a G4-dependent fashion, r(GGGGCC)$_6$ NMM staining experiment−r(GGGGCC)$_6$ was annealed under mG4-forming conditions (see methods). It was then subjected to NMM staining prior to acquisition. The image is split to show the brightfield channel (on the right) and the NMM fluorescence channel (on the left). 10 μm scalebar. **b** r(GGGGCC)$_6$ NMM photo-oxidation experiment−Upon laser excitation the NMM dye bound to the mG4s photo-oxidises the guanine in the quadruplex, leading to the disassembly of the structure (see methods). The region of interest (ROI) highlighted in the figure was irradiated for 1 min and the second image was subsequently acquired, showing condensate disassembly due to photo-oxidation. 10 μm scalebar. Micrographs from panels **a** and **b** are representative of n = 2 respective independent replicates. **c** Brightfield and NMM-stained imaging of TDP-43 RRM1−2 and (GGGGCC)$_{11}$− (GGGGCC)$_{11}$ was annealed under mG4-forming conditions (1−300 μM) and then incubated with 10 μM TDP-43 RRM1−2, revealing a synergist aggregation effect between the (GGGGCC)$n$ repeats and TDP-43. 100 μm scalebar. Micrographs in panel **c** are representative of 3 technical replicates and $n$ = 2 independent biological replicates.

perhaps the protein could act in conjunction with G4-triggered aggregation in a live cell environment, which we validated in this study.

Notably, the condensates responded to staining with the G4-specific dye NMM and were successfully disassembled via guanine photo-oxidation when NMM was irradiated, which could be further explored as an approach to disassemble the pathological aggregates in live cells. As a proof of concept, the biological relevance of our study was assessed by testing the equivalent RNA (GGGGCC)$_6$ sequence, successfully confirming the G4-driven aggregation behaviour also when the RNA repeats were used.

Furthermore, we demonstrated how the G4-ligand PDS can diminish mG4-induced aggregation of (GGGGCC)$_{10}$ by driving the system to a lower molecularity state, favouring the formation of unimolecular, rather than multimolecular G4s. In previously reported

studies, G4-ligands have been shown to prevent aggregation in ALS/FTD disease models[22,69]. It was hypothesised that the ligands interfere with the downstream biological functions of the gene, specifically in relationship to the toxic translated protein[48], or displace proteins bound to the repeats. Although these pathways are still plausible, here we offer additional insights into the aggregation properties of such repeat expansions in the absence of proteins, demonstrating that PDS could also be affecting disease phenotypes in this context by shifting G4 towards lower molecularity species, thus preventing mG4-driven aggregation. However, it is noteworthy that previous studies report conflicting relationships between the frequency of RNA *foci* and age of disease onset[70,71] and no correlation between RNA *foci* aggregation and cognitive decline[72], suggesting that the relationship between RNA *foci* and the clinical manifestation of *C9orf72* ALS may be nuanced, context-

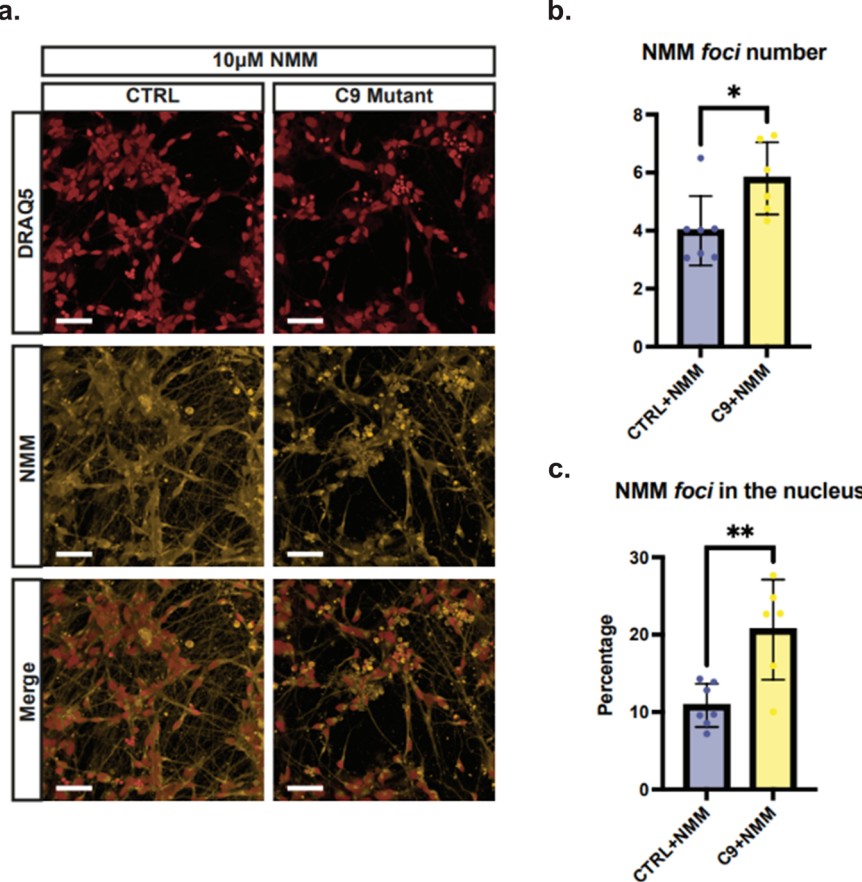

**Fig. 7 | *C9orf72* mutant iPSC-derived motor neurons exhibit an increased number of G4 *foci* per cell soma and increased percentage of G4 *foci* within the nucleus, when compared to healthy controls. a** Confocal representative images of iPSC-derived motor neurons labelled with nuclear stain DRAQ5 (red) and NMM (yellow). Upon staining with NMM, both control and *C9orf72* mutant cells exhibit fluorescent G4 *foci*. 50 μm scale bar. **b** Quantification of the number of NMM *foci* per cell soma in control and mutant motor neurons. *C9orf72* mutant cells exhibit a significant increase in the number of NMM *foci* per cell. *P*-value = 0.0140 from two-tailed Mann-Whitney *U* test. **c** Quantification of the percentage of NMM *foci* in the nucleus in control and mutant motor neurons. Mutant cells exhibit a significant increase in the percentage of NMM *foci* within the nucleus. *P*-value = 0.0038 from unpaired two-tailed *t* test. Data for panels **b** and **c** are from $n = 2$ healthy and $n = 2$ mutant cell lines examined over four repeats with nine fields of view per repeat and presented as mean ± SD, *$p < 0.05$, **$p < 0.01$.

specific, and perhaps less linked to cognitive function. Although our data show the potential relevance of mG4s in *C9orf72*, further work needs to be performed to tackle the underlying mechanism of pathological progression.

To further demonstrate the physiological relevance of our findings, we then incubated our mG4-based aggregates in the presence of the most widely recognised pathologically relevant protein in ALS/FTD (TDP-43). We revealed that both the protein and the nucleic acid aggregation properties are enhanced when incubated together. Furthermore, we confirmed the physiological relevance of our observation, by successfully staining *C9orf72* mutation carrying patient-derived iPSC-motor neurons with NMM and comparing to healthy motor neurons. NMM staining revealed increased prevalence of G4-structures in *C9orf72* mutant motor neurons compared to control counterparts, suggesting that the model proposed in this study could be of both pathophysiological and therapeutic relevance.

In summary, our data indicate a protein-free pathway for the formation of RNA condensates that could be relevant for the initiation and/or progression of *C9orf72* mutation-related ALS and FTD pathogenesis, making mG4s a potential therapeutic target for these devastating and currently incurable neurodegenerative diseases. Importantly, this does not exclude the relevance of proteins or other canonical and non-canonical nucleic acid structures as contributors to these aggregates, as indicated by the condensation synergy observed in the presence of TDP-43.

Currently, the only three FDA-approved drugs that have shown a modest increase in survival rate in randomised clinical trials across the majority of ALS cases are Riluzole, Edaravone and AMX0035[73–76] but the need for clinically impactful therapies is apparent and urgent. Other trials targeting specific familial causes of ALS include a promising GAPmer antisense oligonucleotide (ASO) for SOD1 mutation carrying patients (approximately 2% of all ALS cases) recently being approved by the FDA[77]. However, the first *C9orf72* ASO (BIIB078) trial was discontinued by Biogen for failure to meet its secondary endpoints (NCT03626012). Although further trials using different and possibly more effective *C9orf72* ASOs are underway, it is crucial to consider emerging insights into the disease mechanisms to maximise our repertoire for therapeutic strategy. So far, candidate drugs have been developed to target protein–led aggregation but have shown only partial effectiveness[78], suggesting the need to explore alternative approaches. A recent study investigated a small molecule with the ability to cross the blood-brain barrier and bind to r(GGGGCC)$_n$ promoting formation of a hairpin structure instead of G4s[23], which demonstrates how perturbing the RNA-structural dynamics of the (GGGGCC)$_n$ repeat is a potentially viable strategy for therapeutic intervention. Altogether, our findings suggest that targeting multimolecular RNA G4-structures within the (GGGGCC)$_n$ repeat represents an exciting and potentially tractable strategy for treating ALS/FTD and merits further investigation.

## Methods

### (GGGGCC)$_n$ mG4s annealing protocol

(GGGGCC)$_n$ ($n$ = 1–12) was purchased from IDT in its lyophilised form and diluted in MilliQ water to reach a stock concentration of 1 mM. The stocks were heated to 40 °C for 5 min to ensure complete solvation. The DNA samples were then further diluted to the desired annealing concentration (500–0 µM) in 10 mM TRIS-HCl buffer at pH 7.4 (filtered with 0.2 µm sterile syringe filters) and 30% poly(ethylene glycol) BioUltra 200 (PEG) was added to the clear solution. The samples were vortexed and subjected to a first denaturing step in a C1000 Touch™ Thermal Cycler (heating to 95 °C for 1 h followed by cooling to 25 °C with a temperature ramp of 0.25 °C/min). 500 mM KCl was added to the denatured solution and a further heating cycle was applied (heating to 95 °C for 1 h followed by cooling to 25 °C with a temperature ramp of 0.02 °C/min). To note is that after the long annealing procedure, significant evaporation is observed in the samples. The phase diagram was developed by reducing evaporation as much as possible to estimate an accurate concentration. For this purpose, the author recommends using an annealing volume of > 30 µL. For other applications where exact concentration is not of interest smaller volumes can also be adopted.

Annealing in absence of PEG was performed in the same way but with a slower cooling rate (heating to 95 °C for 1 h followed by cooling to 25 °C with a temperature ramp of 0.01 °C/min).

### Circular Dichroism spectroscopy protocol

Pre-annealed (GGGGCC)$_n$ ($n$ = 1-12) DNA samples were diluted to 5 µM in 10 mM TRIS-HCl buffer pH 7.4 with a final volume of 150 µL. The samples were transferred in Quartz High Precision cell (1 mm light path) and analysed in the Jasco J-715 Spectropolarimeter. The parameters used for each run were the following: Sensitivity (100 mdeg); Wavelength start and end (320 nm–220 nm); Data Pitch 0.5 nm with continuous scanning mode and scanning speed of 100 nm/min; Response (2 s); Band Width (2.0 nm); Accumulation (3).

### Agarose gel electrophoresis protocol

**SYBR safe staining.** Pre-annealed (GGGGCC)$_n$ ($n$ = 1–12) DNA samples were diluted to 10 µM in 10 mM TRIS-HCl buffer pH 7.4 with a final volume of 12 µL. 2 µL of Purple Dye were added to the solution to visualise the samples. The agarose gel matrix was prepared by mixing 1.5 g of agarose in 50 mL 1X TBE buffer and 3 µL SYBR safe stain, heated by microwaving and left to polymerise for 20 min prior to running. The samples were run at 65 V for 85 min at room temperature or 50 V for 150 min on ice (depending on the application) and imaged with Typhoon FLA 9500 on the EtBr channel. Running the gel on ice for longer times and at a lower voltage makes it easier to distinguish multimolecular species.

**NMM staining.** Pre-annealed (GGGGCC)$_n$ ($n$ = 1–12) DNA samples were diluted to 10 µM in 10 mM TRIS-HCl buffer pH 7.4 with a final volume of 12 µL. 2 µL of 50% glycerol were added to the solution to visualise the samples. The agarose gel matrix was prepared by mixing 1.5 g of agarose in 50 µL 1X TBE buffer, heated by microwaving and left to polymerise for 20 min prior to running. The samples were run at 65 V for 85 min at room temperature or 50 V for 150 min on ice (depending on the application). The gel was then incubated in a solution of 50 µL 2 mM NMM and 50 mL 1X TBE for 1 h covered to avoid photobleaching of the dye. The gel was washed once with 1X TBE for 10 min and imaged with Typhoon FLA 9500 on the EtBr channel. Running the gel on ice for longer times and at a lower voltage makes it easier to distinguish multimolecular species. Notably, due to the overlapping emission/excitation range of NMM and SYBR safe, both stainings cannot be performed on the same gel.

## Imaging via confocal microscopy

**Phase diagram.** Leica SP8 Inverted Confocal microscope (HC PL APO CS2 10x/0.40 DRY | HC PL APO CS2 20x/0.75 DRY objectives) was used for the imaging. In order to define aggregation on the phase diagram, pre-annealed (GGGGCC)$_n$ ($n$ = 1–12) DNA samples were directly transferred in PDMS wells and imaged by irradiating the sample with the 458 nm excitation laser on the brightfield channel.

**FAM-labelled samples.** Leica SP8 Inverted Confocal microscope (HC PL APO CS2 10x/0.40 DRY | HC PL APO CS2 20x/0.75 DRY objectives) was used for the imaging. Pre-annealed samples labelled with 2–50% FAM were diluted to 10 µM in 10 mM TRIS-HCl buffer pH 7.4, imaged by irradiating with the 488 nm laser and with an emission filter from 510–550 nm.

**NMM-stained samples.** Leica SP8 Inverted Confocal microscope (HC PL APO CS2 10x/0.40 DRY | HC PL APO CS2 20x/0.75 DRY objectives) was used for the imaging. Pre-annealed samples were diluted to 10 µM in 10 mM TRIS-HCL buffer pH 7.4 and irradiated with the 405 nm laser. 10 µM NMM was then added to the sample and left to incubate for 10 min in a dark environment. The sample was then imaged on the emission filter 600-650 nm.

**G4-ligands containing samples.** Leica SP8 Inverted Confocal microscope (HC PL APO CS2 10x/0.40 DRY | HC PL APO CS2 20x/0.75 DRY objectives) was used for the imaging. Pre-annealed (GGGGCC)$_{10}$ in presence of the 10 µM PDS was directly transferred in PDMS wells and imaged by irradiating the sample with the 458 nm excitation laser on the brightfield channel. As PDS is solubilised in presence of 0.1% DMSO, a DMSO control was also added to the experiment.

**RRM1–2 containing samples.** Leica SP8 Inverted Confocal microscope (HC PL APO CS2 10x/0.40 DRY | HC PL APO CS2 20x/0.75 DRY objectives) was used for the imaging. Pre-annealed (GGGGCC)$_{11}$ in presence of the 10 µM RRM1-2 was directly transferred in PDMS wells and imaged by irradiating the sample with the 405 nm excitation laser on the brightfield channel. NMM was then added in a 1:1 NMM:DNA ratio and left to incubate for 10 min. The sample was then imaged again on the emission filter 600–650 nm.

**(GGGGCC)$_{11}$-FAM and RRM1–2-ALEXA633 samples.** Leica SP8 Inverted Confocal microscope (HC PL APO CS2 10x/0.40 DRY | HC PL APO CS2 20x/0.75 DRY objectives) was used for the imaging. Pre-annealed (GGGGCC)$_{11}$ (10% fluorescently labelled with FAM) in presence of the 10 µM RRM1-2 (fluorescently labelled with ALEXA633) was directly transferred in PDMS wells and firstly imaged by irradiating the sample with the 405 nm excitation laser on the brightfield channel. The DNA component was imaged by irradiating with the 488 nm laser and with an emission filter from 510–550 nm. The protein component was imaged by irradiating with the 633 nm laser and with an emission filter 610–700 nm).

**NMM-photooxidation.** Leica SP8 Inverted Confocal microscope (HC PL APO CS2 10x/0.40 DRY | HC PL APO CS2 20x/0.75 DRY objectives) was used for the imaging. Pre-annealed (GGGGCC)$_6$ sample was diluted to approximately 10 µM in 10 mM TRIS-HCL buffer pH 7.4 and irradiated with the 405 nm laser. 10 µM NMM was then added to the sample and left to incubate for 10 min in a dark environment. A portion of the sample was then irradiated at 405 nm at 100% for 1 min and complete disassembly of that portion of the aggregate was observed both in brightfield or on the emission filter 600–650 nm.

**FRAP (Fluorescence recovery after photobleaching).** Leica SP8 Inverted Confocal microscope (HC PL APO CS2 10x/0.40 DRY | HC PL APO CS2 20x/0.75 DRY objectives) was used for the imaging. Pre-

annealed samples labelled with 10% FAM were imaged by irradiating with the 488 nm laser and with an emission filter from 510–550 nm. A region of the well was irradiated at 100% power intensity for 30 s until no fluorescence was observed. Recovery after photobleaching was monitored every 5 min for 30 min.

## dsDNA condensates annealing protocol

Condensates made from dsDNA nanostars were prepared by mixing four nanostar core strands (N4_core1, N4_core2, N4_core3, N4_core4) and two linker strands (L_AA1 and L_AA2) in a TrisHCl buffer containing 500 mM KCl at a final nanostar concentration of 1 µM and linkers concentration of 2 µM. A volume of 70 µL was then loaded into borosilicate glass capillaries (inner dimensions: 0.4 mm × 4 mm × 50 mm, CM Scientific) with a micropipette. Prior to loading, capillaries were cleaned by sonication in 1% Hellmanex III (HellmaAnalytics) at 60 °C for 15 min. The surfactant was removed through five rounds of rinsing with deionised (DI) water, followed by a round of sonication in ultrapure water (Milli-Q). After loading the sample, the capillary ends were capped with mineral oil and sealed with epoxy glue (Araldite) on a coverslip. The sample was annealed by incubating at 95 °C for 30 min and cooled down between 85 °C and 25 °C (−0.1 °C/min cooling rate). Samples were transferred from the glass capillary to a PDMS well for imaging. All strands were purchased from Integrated DNA Technologies (IDT) and reconstituted in TE. Sequences are provided in the SI, Fig. S4A.

## (GGGGCC)$_n$ mG4s annealing protocol in presence of PDS

(GGGGCC)$_{10}$ was purchased from IDT in its lyophilised form and diluted in MilliQ water to reach a stock concentration of 1 mM. The stock was heated to 40 °C for 5 min to ensure complete solvation. The DNA sample was then further diluted to the desired annealing concentration (150 µM) in 10 mM TRIS-HCl buffer at pH 7.4 (filtered with 0.2 µm sterile syringe filters) and 30% poly(ethylene glycol) BioUltra 200 (PEG) was added to the clear solution. The samples were vortexed and subjected to a first denaturing step in a C1000 Touch™ Thermal Cycler (heating to 95 °C for 1 h followed by cooling to 25 °C with a temperature ramp of 0.25 °C/min). 500 mM KCl and 10 µM PDS were added to the denatured solution and a further heating cycle was applied (heating to 95 °C for 1 h followed by cooling to 25 °C with a temperature ramp of 0.02 °C/min). To solubilise the PDS, 0.1% DMSO was also added to the solution, therefore a DMSO control has also been added to the results. To note is that after the long annealing procedure, significant evaporation is observed in the samples. The phase diagram was developed by reducing evaporation as much as possible to estimate an accurate concentration. For this purpose, the author recommends using an annealing volume of >30 µL.

## (GGGGCC)$_{11}$ and RRM1–2 aggregates

**RRM1–2 expression and purification.** Purification of the TDP-43 RRM1-2 fragment was carried out as previously described[53]. Briefly, the protein construct was expressed in a pET-SUMO expression vector containing the kanamycin antibiotic resistance gene. The plasmid was expressed by heat-shock transformation in BL21 *Escherichia coli* (*E. Coli*) as a protein fused with a SUMO solubilisation tag and a 6×His tag. Cells were grown in Luria-Bertani (LB) with 50 µg/mL kanamycin at 37 °C with 200 rpm shaking until they reached an OD of 0.7 at 600 nm. Protein expression was induced with 0.5 mM IPTG, and cells were grown overnight at 18 °C with 200 rpm shaking. Cells were then collected by centrifugation at 4000 rcf for 20 min at 4 °C and resuspended in lysis buffer (10 mM potassium phosphate buffer pH 7.2, 150 mM KCl, 5 mM imidazole, 5% v/v glycerol, 1 mg/mL lysozyme, complete™ EDTA-free Protease Inhibitor tablet by Roche, 1 µg/mL DNase I and 1 µg/mL RNaseA). The resuspended cells were sonicated on ice for 20 min (15 s on and 45 s off pulses, 20% amplitude) and soluble proteins were recovered by centrifugation at 18,000 rcf for

45 min at 4 °C. The supernatant was filtered using a 0.22 µm filter and loaded onto a HisTrap HP 5 ml column (Cytiva) which was pre-equilibrated with 10 mM potassium phosphate buffer pH 7.2 supplemented with 15 mM imidazole (binding buffer). The 6xHis-SUMO-construct was eluted with phosphate buffer supplemented with 300 mM imidazole (elution buffer). After measuring the concentration of RRM1-2 on a Nanodrop (Thermo Fisher Scientific), the eluate was dialysed overnight at 4 °C against phosphate buffer in the presence of Tobacco Etch Virus (TEV) protease (1:10 TEV:protein construct molar ratio) to remove the 6xHis-SUMO tag. A second nickel-affinity chromatography was performed, and the flowthrough was loaded onto a HiTrap Heparin column (Cytvia) which was pre-equilibrated with 10 mM phosphate buffer. The protein was eluted with high-salt phosphate buffer (10 mM potassium phosphate buffer pH 7.2, 1.5 M KCl) and applied on a HiLoad 16/60 Superdex 75 prep grade column (Cytvia) pre-equilibrated with 10 mM phosphate buffer pH 7.2 for size-exclusion chromatography. Protein concentration was taken on a Nanodrop, and protein identity and purity were checked by PAGE and mass spectrometry. All protein purification steps were carried out using an ÄKTA Pure (Cytvia).

**SDS-PAGE.** Samples were prepared in 4X LDS sample buffer and 10X reducing agent and boiled at 95 °C for 5 min. Samples were run on 4–12% Bis-Tris NuPAGE gels (Thermo Fisher Scientific) at 180 V for 35 min. Gels were stained with InstantBlue Coomassie Protein Stain (Abcam) for 15 min, washed in water for 30 min and imaged on an ImageQuant LAS 4000.

**Electrospray ionisation mass spectrometry (ESI-MS).** ESI-MS was performed on purified protein samples (50 µM) buffer-exchanged into HPLC grade water to confirm molecular weight and sample purity. ESI-MS was performed by Malgorzata Puchnarewicz using the Chemistry Mass Spectrometry facilities available at the Molecular Sciences Research Hub, Department of Chemistry, Imperial College London.

**RRM1–2 labelling with ALEXA 633.** Purified RRM1-2 was fluorescently tagged using Alexa Fluor™ 633 C$_5$ Maleimide (Thermo Fisher Scientific). The dye was added to a sample of purified protein (1:10 protein:dye molar ratio) in 10 mM potassium phosphate buffer pH 7.2 and incubated at 25 °C for 4 h on a roller mixer with slow rotation. The solution was then dialysed against 10 mM potassium phosphate buffer pH 7.2 overnight to remove unbound dye.

**DNA/protein aggregates formation.** (GGGGCC)$_{11}$ was annealed at different concentrations (300, 100 and 1 µM as per mG4-annealing protocol). The samples were then incubated for 3 days at 37 °C with 10 µM of TDP-43 RRM1-2.

**Transmission electron microscopy (TEM).** To prepare TEM grids, 10 µL of sample was pipetted on Formvar/Carbon coated 300 mesh copper grids (Agar Scientific) and left to incubate for 2 min. Excess sample was removed with Whatman filter paper. The grids were then washed with dH$_2$O and stained with uranyl acetate (2% w/v) for 2 min. Excess uranyl acetate was removed, and the grid was left to dry for at least 5 min. For each sample, images were taken in at least 5 locations on the grid to ensure consistent structures were present across the grid. Grids were measured on a T12 Spirit electron microscope (Thermo Fisher Scientific (FEI), Hillsboro, OR, USA)

## iPSC derived neuronal cell culture

Experimental protocols were carried out according to approved regulations and guidelines by UCL Hospitals National Hospital for Neurology and Neurosurgery and UCL Institute of Neurology joint research ethics committee (09/0272). iPSCs were maintained with Essential 8 Medium media (Life Technologies) on Geltrex (Life

Technologies) at 37 °C and 5% carbon dioxide. iPSCs were passaged when reaching 70% confluency using EDTA (Life Technologies, 0.5 mM). iPSC cultures underwent differentiation into spinal cord motor neurons as previously described[55]. Briefly, iPSCs were plated to 100% confluency and then differentiated to a spinal neural precursor fate by sequential treatment with small molecules, day 0–7: 1 μM Dorsomorphin (Tocris Bioscience), 2 μM SB431542 (Tocris Bioscience), and 3.3 μM CHIR99021 (Tocris Bioscience), day 7–14: 0.5 μM retinoic acid (Sigma Aldrich) and 1 μM Purmorphamine (Sigma Aldrich), day 14–18: 0.1 μM Purmorphamine. After neural conversion and patterning, 0.1 μM Compound E (Bio-techne) was added for terminal differentiation into spinal cord motor neurons.

**Immunocytochemistry (ICC).** Cells were seeded on clear bottom 96 well plates (PerkinElmer) at a density of 30,000 cells per well. After 24 h, Compound E (Bio-techne) was added for terminal differentiation into spinal cord motor neurons. At day 6 of terminal differentiation, DRAQ5 (Merck) was applied to the cells diluted at 1:1000 in media and incubated for 10 min to stain nuclei. Cells were then fixed in 4% paraformaldehyde for 15–20 min at room temperature, followed by permeabilization using 0.3% Triton-X (NMM and antibody) and blocking with 5% BSA in PBS for 60 min (antibody). Either 10 μM NMM (Cambridge Bioscience) or 0.5% DMSO diluted in 0.15% Triton-X containing PBS, or a primary antibody was then applied. The NMM/DMSO was incubated for 40 h at 4 °C while the primary antibody anti-tubulin beta-III (#802001, BioLegend), diluted at 1:2000 in 0.15% Triton-X in PBS, was left overnight at 4 °C. Plates treated with NMM/DMSO were imaged directly. Secondary antibody incubation was then performed using Alexa Fluor-conjugated secondary antibody (anti-rabbit) at 1:1000 dilution in PBS at room temperature. Cells were imaged with Opera Phenix (PerkinElmer) at 40X, Z series of images were used. Standard imaging and acquisition settings were applied. 9 fields of view were taken for each well. Cell density was measured, and wells with nuclei object number > 150 in all fields were proceeded for analysis. All cell images were analyzed through the Harmony Imaging and Analysis Software (PerkinElmer).

**Flow cytometry.** Motor neurons were dissociated into single cells using Accutase (Life Technologies). The cell suspensions were fixed using 4% paraformaldehyde for 15–20 min and then permeabilised using 0.3% Triton-X in PBS for 1 h at room temperature. Subsequently, they were treated with either 10 μM NMM (Cambridge Bioscience) or 0.5% DMSO diluted in 0.15% Triton-X overnight at 4 °C. The samples (10,000 cells/sample) were examined using a BD LSRFortessa analyser operated by the FACSDiva software (BD), and the results were analysed using the FlowJo software.

### Reporting summary
Further information on research design is available in the Nature Portfolio Reporting Summary linked to this article.

## Data availability
The data generated in this study have been deposited in the Supplementary Information and Source Data file (https://doi.org/10.17863/CAM.104057) University of Cambridge Research Data Facility.

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

## Acknowledgements

Graphics were designed using Procreate design app and ChemDraw. The authors thank the Crick Flow Cytometry and High Throughput Screening STPs for their technical assistance. The authors thank Prof. Annalisa Pastore for providing the expression plasmid for the TDP-43 RRM1-2 protein. This work was supported by the following funding bodies: M.D.A. is supported by Biotechnology and Biological Sciences Research Council (BBSRC) David Phillips Fellowship [BB/R011605/1]; M.D.A. is a Lister Institute Fellow and is supported by a Lister Institute Research Prize. F.R. is supported by a Leverhulme Trust, Leverhulme Cellular Bionics scholarship [EP/S023518/1]; L.D.M. is supported by Royal Society University Research Fellowship [UF160152, URF\R\221009] and Royal Society Research Fellows Enhanced Research Expenses [RF/ERE/210029], also supporting R.R.S. L.D.M., R.R.S., D.T., and L.M. are supported by the European Research Council (ERC) under the Horizon 2020 Research and Innovation Programme [ERC-STG No 851667 NANOCELL]. Y.E. is supported by United Kingdom Research and Innovation Future Leaders Fellowship [MR/S031537/1]. T.E.M. is supported by [EPSRC G98210]. F.A.A. is supported by United Kingdom Research and Innovation Future Leaders Fellowship [MR/S033947/1] and Alzheimer's Research UK [ARUK-PG2019B-020]. This work was also funded by the Francis Crick Institute, which receives its core funding from Cancer Research UK (FC010110), the UK Medical Research Council (FC010110), and the Wellcome Trust (FC010110). R.P. holds an MRC Senior Clinical Fellowship (MR/S006591/1) and a Lister Research Prize Fellowship. A.H. is supported by The Engineering and Physical Research Council (Grant Number: EP/S023518/1). R.B. is an NIHR Academic Clinical Lecturer in Neurology at UCL and is funded by an Academy of Medical Sciences Starter Grant for Clinical Lecturers (SGL027\1022). M.P.H. is funded through an MRC grant (MR/S006591/1). Y.W. is supported by the Motor Neurone Disease Association. Facility for Imaging by Light Microscopy (FILM) at Imperial College London is part-supported by funding from the Wellcome Trust [grant 104931/Z/14/Z] and BBSRC [grant BB/L015129/1]. Funding for open access charge: Imperial College London.

## Author contributions

F.R. developed the experimental protocols and conducted all experiments unless otherwise stated. D.T. designed and produced the dsDNA condensates control. A.H. expressed and purified TDP-43 and contributed to performing the TDP-43 incubation experiments. J.L. conducted the work associated with iPSC derived neuronal cells. Y.W. aided with the imaging and FACS for the NMM staining experiments. T.E.M. synthesised PDS and supported the related experiments. L.M. supported the RNA and DNA photooxidation experiments. R.R.S. helped generating the 3D reconstructed image of the fluorescence condensates. F.A.A. and D.M.V. aided in the design of the TDP-43 experiments. D.M.V. and A.H. performed the TEM experiments. R.B., R.P., M.P.H., Y.W., and M.D.A. contributed to designing the cellular experiments. F.R. analysed all the data with support from M.D.A., L.D.M. and R.P. F.R. and M.D.A. co-wrote the paper, with support from L.D.M. and R.P. M.D.A. and F.R. has designed the overall research design with substantial support by L.D.M. and Y.E. M.D.A., L.D.M. and Y.E. supervised the research. All authors discussed the results and edited the paper.

## Competing interests

The authors declare no competing interests.
