## [Peer Review File · Nature Communications]

The ALS/FTD-related *C9orf72* hexanucleotide repeat expansion forms RNA condensates through multimolecular G-quadruplexesREVIEWER COMMENTS

Reviewer #1 (Remarks to the Author):

In this work, Raguseo and co-authors investigate the aggregation of the hexanucleotide repeat involved in ALS and FTD diseases. The authors ascribe this aggregation to the formation of multimeric G-quadruplexes (mG4s), both from DNA and RNA.

The paper is well written, the idea of the work is good, however in my opinion not all the conclusions are supported by the results. In particular, the authors did not fully demonstrate that what they observed really are mG4s, rather than a kind of aggregation due to other secondary structures.

My major concerns are about the first part of the results. In particular:

- why such a high KCl concentration has been used: 500 mM is far from physiological conditions. While 30% PEG mimics intra-cellular crowding, 500 mM KCl does not.

- Legend in figure 2A is in contrast to methods 2: were oligo "annealed at 500 μ M, 250 μ M, 100 μ M and 50 μ M" or "diluted to 5 μ M"?

- Figure 2B: I do not agree with the authors' interpretation. It's true, there are smears indicating that probably there are higher molecularity G4s, but there should more bands, a sort of ladder where 2, 4 or more G4s fold to form mG4s (see, as an example <https://doi.org/10.1074/jbc.C113.452532>, Fig. 1I). The smears do not support the hypothesis of mG4s formation, there could be other secondary structures, different from mG4s. Polyacrylamide gels, with a higher resolution, better with fluorescent- or radioactive-labeled oligonucleotides, should be provided to answer this point.

- Figure 2B: why no band can be seen in the (GGGGCC)₂ lane? At least the unfolded form should be visualized.

- Figure 2C: why do lower bands (in the green box) migrate higher when KCl increases? They should be "Low/No G4-content species" therefore "unaffected by K⁺ ions". Is that migration due to the so high KCl concentration? Maybe it is aspecific binding of K⁺ ions to DNA, thus neutralizing the overall negative DNA charge? This gel does not support the hypothesis of mG4s formation.

- Figure 2D: NMM stains a low band in the (GGGGCC)₈ lane, probably corresponding to the lower band (green box) in Figure 2B, same lane. If mG4s form, that band should migrate slowly. This seems to be (again) in contrast with the formation of mG4s: please comment about it.

In general Figure 2 does not fully support the hypothesis of mG4s formation, the authors need to perform polyacrylamide gels and other experiments to clarify this point, for example mass spectrometry under mild conditions.

Results reported in Figure 3 could be ascribed both to the formation of mG4s but also to oligonucleotides aggregation, containing G4s but not multimeric G4s.

Reviewer #2 (Remarks to the Author):

The manuscript from Di Antonio and co-workers studies a very timely and interesting topic. ALS and FTD, two neurodegenerative diseases, are caused by a repeat expansion (> 24, can be thousands) of an hexanucleotide sequence in the non-coding region (5' UTR) of C9ORF72. The expanded GGGGCC impairs normal transcription of C9ORF72 leading to reduced C9ORF72 mRNA and protein in the frontal cortex and spinal cord of patients.

Here in the presented manuscript, they analyzed if alternative nucleic acid structures can form in these repeats and which contribution, they have on the formation of repeat induced aggregates. Based on their work they conclude that G-quadruplexes (G4s) contribute to this aggregation process. They used a protein free in vitro system to monitor aggregation of G-rich oligos and experimental systems. The manuscript is well written but I there are major issues that needs to be addressed.

Major comments:

- The current manuscript has a highly relevant topic, however as a major comment I have is that it lacks an in vivo relevance due to pure in vitro experiments. Moreover, some in vivo aspects are required to strengthen their data and proof that in living cells also high concentrations of G-rich

repeats leads to mG4s and that this drives aggregates. For example, can aggregates be resolved by overexpressing G4 unwinding helicases? Alternatively, can by a G4 unfolding ligand these aggregates be destroyed, similar to PDS? Do proteins bind to these aggregates and drive even further aggregation? Do other alternative structures also form in these aggregates (e.g. hairpins)?

- Are G4s forming in aggregates, or are G-rich oligos assembly?
- Would it be possible to map long range interactions in these constructs to strengthen that really these interactions are mediated by G-G base pairing? Alternatively can HiC or an alternative be done in ALS or FTD cells to monitor similar phenotypes as the pure in vitro system?
- Other repeat expansion disease has strong phenotypes, not all of these repeats have G-rich repeats? Is this specific for ALS and FTD repeats, control repeats or additional repeats needs to be studied?

specific major comments:

- The data presented in Figure 2 are clear but needs to be improved: the quality of the gel images needs to be greatly improved. Native PAGE should be utilized instead of agarose gels. A loading control should be included, or the ssDNA should be visualized to account for the presence of non-G4 bands. Noteworthy, increasing the salt concentration within a gel ("AGE") especially from 0 to 0.5 M already leads to a smile/shift of the samples; how do the authors account for this? Is the NMM and SYBR Safe (not "SYBRSAFE.") data from the same gel; this information the authors do not include. Also, the ladder in Fig. 1C is not labeled, which makes it hard to compare 1B and 1C. Also, a ladder should be included in 1D to make it possible to compare the images. Also, greater care should be taken in the methods section where it states 1.5 g agarose in 50 μ l TBE. I guess, the authors used a 50 ml mini-gel instead.

- In line 182 the statements "Having demonstrated the formation of multimolecular constructs" is a little bit of an over-interpretation, as the data in figure2 only show the presence of multiple different populations of folded oligodeoxynucleotides.

- Which impact does crowding has on these events, So far the authors have used PEG to promote phase separation. Have they considered to monitor other methods (e.g. by pressure, crowing peptides, Ficoll, dextran) to induce crowing? Is the effect only due to crowing, as other molecules also have CD changes in crowing conditions and DNA dynamic change in crowing conditions, (e.g. Harve et al 2008, Mardoum et al 2018).

- Cloning and working with repeats of Guanine rich stretches is very difficult and the authors have established very good working methods for these repeats. However, it is essential to control the length of the repeats prior each experiment. I also recommend increasing the repeat size from n-24 to at least double or even higher to resemble the in vivo situation.

- Although the data are very interesting unfortunately the quality of Fig 3A is not sufficient to observe nucleic acid structure. Given the size bars in Fig. 3B and C the structure which is clearly visible in Fig. 3B and C should be approximately half as big in Fig. 1A. Perhaps a second panel, in which a stain was used similar to Fig. 3B,C would be helpful to actually visualize the structures in Fig. 3A.

- Do the authors have any prove that NMM is not stabilizing, supporting these structures as NMM has been published as a G4 ligand?

- In the PDS studied they showed that PDS can diminish mG4. As these data contrast with previous work and very interesting and strong. I recommend performing at least 2 additional G4 ligands in these assays. They suggest in the discussion the G4-G4 binding interaction is disturbed by the ligand. As they do not have proteins in their systems, it is tempting to speculate that the G4 itself is causing this effect and the ligand will prevent aggregation this opens a potential for new therapeutic applications. This speculation is very logic and elegant but based on the current data set too speculative and needs experimental prove. Have the authored considered that in the aggregates G-rich regions cluster together and do not form a G4?

- Figure 4A, while panel 4C and b is of good quality and gel in D and E are not sufficient. there is no clear image visible for GGGTT and CCCC GG, also why do the scale bars differ in length. Here guidance in the text and improve figures are required to understand these data.

Reviewer 1

1) Why such a high KCl concentration has been used: 500 mM is far from physiological conditions while 30% PEG mimics intra-cellular crowding, 500 mM KCl does not.

We agree with the reviewer that 500 mM KCl does not mimic physiological conditions. We used high K^+ concentration to enhance mG4-formation. To address this concern, we have generated new data revealing that aggregation can still be observed at 100 mM KCl for repeat length $(GGGGCC)_{11}$. We have now added this new evidence in the supporting information of the revised manuscript (**Figure S5**) and attached it here for convenience.

Figure S5: DNA $(GGGGCC)_{11}$ aggregates form at physiological concentrations of KCl. Brightfield imaging in of 500 μ M $(GGGGCC)_{11}$ annealed with 100/300/500 mM KCl. 50 μ m scalebar.

2) Legend in figure 2A is in contrast to methods 2: were oligo “annealed at 500 μ M, 250 μ M, 100 μ M and 50 μ M” or “diluted to 5 μ M”?

The samples were annealed at the different concentrations stated (respectively 500 μ M, 250 μ M, 100 μ M and 50 μ M) and successively diluted at 5 μ M to perform Circular Dichroism. We have now edited the figure caption to clarify this.

3) Figure 2B: I do not agree with the authors’ interpretation. It’s true, there are smears indicating that probably there are higher molecularity G4s, but there should more bands, a sort of ladder where 2, 4 or more G4s fold to form mG4s (see, as an example <https://doi.org/10.1074/jbc.C113.452532>, Fig. 1I). The smears do not support the hypothesis of mG4s formation, there could be other secondary structures, different from mG4s. Polyacrylamide gels, with a higher resolution, better with fluorescent- or radioactive-labelled oligonucleotides, should be provided to answer this point.

Whilst we agree with the reviewer that smearing is not conclusive to demonstrate formation of multimolecular G4s, the use of the GGGGTT mutant that forms stable unimolecular G4s¹ but not multimolecular ones (**Figure 4**), indicates that to observe aggregation multimolecular G4s needs to be formed. Nevertheless, we agree with the reviewer that a gel displaying clearer species would be ideal to further illustrate mG4-formation. To achieve this, we have performed

an additional agarose gel by annealing the FAM-labelled oligos at 1 μM (lower concentration than before) (**Figure S1A**) to reduce smearing and to visualise more clearly multimeric species, keeping consistency with the type of gels used in the manuscript (i.e. agarose rather than PAGE). **Figure S1A** indicates that under these conditions we can distinguish more clearly multimeric G4-species (bi-molecular and tetra-molecular) populations formed by $(\text{GGGGCC})_n$ in a length dependent fashion, similarly to what described in previous reports mentioned by the reviewer. We have now added this new evidence in the SI and discussed it in the main manuscript, which we also attached here for convenience.

Figure S1A: FAM labelled $(\text{GGGGCC})_n$ AGE – FAM labelled $(\text{GGGGCC})_n$ was annealed at 1 μM under G4-forming conditions (KCl 500 mM). Three distinct bands can be detected, respectively corresponding to bi-molecular and tetra-molecular species, with higher molecular weight species formed at higher repeat lengths, while lower molecular weight species are more prominent at lower repeat lengths.

Revised manuscript: “Although the gel resolution in **Figure 2B** does not allow to visualise distinct bands at higher molecular weight, two main bands can be observed that could be ascribed to a unimolecular and a bimolecular species based on previous literature^{2 3}. These bands can be better visualised by using fluorescently labelled DNA (FAM) at lower concentrations to avoid smearing, as shown in **Figure S1**.”

4) Figure 2B: why no band can be seen in the $(\text{GGGGCC})_2$ lane? At least the unfolded form should be visualized.

The faint staining is due to the use of SYBR safe, which is not particularly efficient in staining short ssDNA. We have now performed a gel using fluorescently labelled oligonucleotides (**Figure S1**) which allows for clear identification of bands also for the short $n = 2$ repeat, as pointed out by the reviewer.

5) Figure 2C: why do lower bands (in the green box) migrate higher when KCl increases? They should be “Low/No G4-content species” therefore “unaffected by K^+ ions”. Is that migration due to the so high KCl concentration? Maybe it is specific binding of K^+ ions to DNA, thus neutralizing the overall negative DNA charge? This gel does not support the hypothesis of mG4s formation.

The reviewer raises a good point here and we have now performed additional experiments to clarify this. We have repeated this experiment using FAM-labelled DNA rather than using SYBR Safe staining that is not efficient for visualising G4 structures. The data displayed in the revised **Figure 2C** (also attached below), still show a K⁺-dependent reduction of electrophoretic mobility of both the low and high molecular weight species, but now reveal a full transition towards the slower migrating band at increasing K⁺ concentrations. To further demonstrate that the slow-moving band could be ascribed to a multi-molecular G4, we have used NMM staining that fluoresces only upon G4-binding. The NMM-stained gel in **Figure 2C** not only demonstrates that the slow-moving bands contain G4s, but also shows stronger fluorescence of fast moving band at higher K⁺ concentrations, suggesting that a monomolecular G4-structure can be formed under these conditions. This trend in NMM fluorescence suggests an increasing prominence of G4 species with increasing [K⁺], which could underpin the trend in electrophoretic mobility noted by the reviewer. We have now added this additional evidence and a short discussion on these observations in the revised manuscript.

Figure 2C: (GGGGCC)₁₁ agarose gel electrophoresis - (GGGGCC)₁₁ was annealed in the presence of different KCl concentrations. As KCl increases, so does the molecular weight of the species involved, indicating that the formed multimolecular structures have a strong KCl dependence. In the schematics, the green ball represents the K⁺ ion. On the right, the samples are stained with NMM, while on the left the sequences are FAM labelled. For reference a 100 base pairs ladder has also been added. For visualisation, the ladder was stained with SYBR Safe.

Revised manuscript: “AGE experiments revealed that increasing K⁺ concentrations can convert the fast-running band, which we ascribed to the unimolecular G4, to the slowly running dimeric band, which is consistent with the formation of a bimolecular G4-structures (**Figure 2C**). ... To corroborate this observation, the AGE experiment was repeated using NMM staining, which confirmed the presence of G4s in the slow-moving band. The intensity of NMM fluorescence recorded in the slow-moving bands increases with K⁺, consistent with a greater abundance of (multimolecular) G4 species (**Figure 2C**).”

6) Figure 2D: NMM stains a low band in the (GGGGCC)₈ lane, probably corresponding to the lower band (green box) in Figure 2B, same lane. If mG4s form, that band should migrate slowly. This seems to be (again) in contrast with the formation of mG4s: please comment about it.

As discussed above, the initial use of SYBR safe can bias electrophoresis observations by under-staining bands with high G4-content. To address this, we have also repeated this gel using fluorescently labelled DNA (see Figure S1B), showing distinct formation of higher molecular weight species for repeat 8. The smearing bands have also been shown to stain for NMM in the same conditions in Figure 2D. This is in agreement with formation of mG4s.

Figure S1B. (GGGGCC)_n FAM labelled AGE – FAM labelled (GGGGCC)_n was annealed at 350 μM under G4-forming conditions (KCl 500 mM). No distinct bands can be observed due to sample aggregation but is evident how higher repeat lengths lead to higher molecular weight species.

Figure 2D. (GGGGCC)_n agarose gel electrophoresis – NMM stain. (GGGGCC)_n ($n = 2-11$) were annealed at 250 μM in mG4-forming conditions. The lanes containing any of the (GGGGCC)_n were fluorescent upon staining with NMM, implying the presence of G4s in the higher molecular weight species formed under these conditions. The absence of staining for both the ladder and the non-G4 ssDNA control confirm the specificity of the dye for G4-containing species.

7) In general Figure 2 does not fully support the hypothesis of mG4s formation, the authors need to perform polyacrylamide gels and other experiments to clarify this point, for example mass spectrometry under mild conditions.

Results reported in Figure 3 could be ascribed both to the formation of mG4s but also to oligonucleotides aggregation, containing G4s but not multimeric G4s.

To address this comment, we have now used a fluorescently labelled DNA as suggested by the reviewer that reveals formation of dimers and tetramers. Besides our responses to point 3-6 above, in our initial submission we also reported further control experiments confirming the role of mG4 in the aggregation pathway we propose. As displayed in **Figure 4**, the $(GGGGTT)_n$ control sequence still forms stable unimolecular G4s (as it would be expected¹) but is less prone to form multimolecular species. This sequence leads to no detectable aggregation under conditions in which $(GGGGCC)_n$ aggregates, demonstrating that unimolecular G4s alone are insufficient to induce aggregation, and that mG4s are required, which we believe addresses in full the reviewer concern. We also ruled out the possibility of base pairing being the cause of the lack of aggregation in $(GGGGTT)_n$ compared to $(GGGGCC)_n$ by using an additional control sequence $(GGGGAA)_n$, which is forming multimolecular G4 species but cannot form G-C base pairs. Furthermore, the hypothesis that mG4s are the prevalent species holding the aggregates together was confirmed by the G4-specific disassembly photo-oxidation experiment in **Figure 3**. Taken together, the combined evidence gathered with both microscopy and gel electrophoresis, testing the effect of systematic sequence variations and potassium concentration, strongly supports our proposed mechanism for mG4-based aggregation.

Addressing Reviewer 2

8) The current manuscript has a highly relevant topic, however as a major comment I have is that it lacks an in vivo relevance due to pure in vitro experiments. Moreover, some in vivo aspects are required to strengthen their data and proof that in living cells also high concentrations of G-rich repeats leads to mG4s and that this drives aggregates. For example, can aggregates be resolved by overexpressing G4 unwinding helicases? Alternatively, can by a G4 unfolding ligand these aggregates be destroyed, similar to PDS? Do proteins bind to these aggregates and drive even further aggregation? Do other alternative structures also form in these aggregates (e.g. hairpins)?

We agree with the reviewer's comment on the limited biological relevance of our initial set of experiments, despite the precise insights they provide on the biophysical mechanisms involved. To address this concern, we have now performed additional experiments to demonstrate the physiological relevance of our findings to ALS/FTD. Firstly, as suggested by the reviewer, we have investigated how the presence of a protein relevant to the disease (TDP-43) affects the aggregation behaviour (**Figure 6C** in the revised manuscript). Interestingly, we have discovered that mG4-structures formed by the $(GGGGCC)_{11}$ repeats can promote TDP-43 aggregation under conditions where no aggregation is observed with neither the protein nor the nucleic acids alone, suggesting a synergistic effect.

Moreover, we have used NMM (that fluoresces only upon binding to G4-structures) to stain iPSC patient-derived neuronal cells, in collaboration with Prof Rickie Patani. The additional data generated revealed that NMM staining can be observed in the RNA-aggregate foci typical of ALS patients (**Figure 7** in the revised manuscript), strongly suggesting that our model of RNA aggregation can be relevant for the formation of aggregates in ALS/FTD patients.

Finally, we would like to point out that the NMM-photo-oxidation experiments reported in **Figure 3** of the manuscript, clearly demonstrate how G4-destabilisation by small molecule treatment causes aggregate disassembly, as indicated by the reviewer. We have attached this new evidence here for convenience.

Figure 6C: Brightfield imaging of TDP-43 RRM1–2 and (GGGGCC)₁₁ NMM-stained aggregates - (GGGGCC)₁₁ was annealed under mG4-forming conditions (1-300 μM) and then incubated with 10 μM TDP-43 RRM1–2. Upon incubation with the protein, the mG4 aggregates trigger protein, indicating a synergist aggregation effect between the (GGGGCC)_n repeats and TDP-43. 100 μm scalebar.

Figure 7: C9orf72 mutant iPSC patient derived spinal motor neurons show efficient fluorescent staining of RNA aggregates via the G4-binder NMM. A. Confocal images of IPSC patient derived neurons. The micrographs show the images taken by overlapping the Nuclear Stain DRAQ5 (in red) and the NMM channel (in yellow). In the DMSO control, no autofluorescence in cells can be observed in the NMM emission channel. Upon staining with the dye, the C9orf72 expansion containing neurons present fluorescent putative RNA aggregates. In the panel on the left, cells were stained with DRAQ5 and the neuronal marker β -tubulin III (in green). **B. Fluorescence-activated cell sorting (FACS) analysis for the C9-mutant motor neurons.** Cells treated with DMSO showed no characteristic fluorescence in the NMM-emission channel via FACS. NMM treatment resulted in detection of fluorescence of the characteristic NMM spectrum, indicating efficient incorporation into motor neurons.

9) Are G4s forming in aggregates, or are G-rich oligos assembly?

Beside the NMM photo-oxidation experiments mentioned in point 8 above, which demonstrate how G4-structures are essential to form the aggregate, we have also performed control experiments in **Figure 4** using G-rich sequences that are less prone to form multimolecular G4-structures, leading to no observable aggregation and demonstrating the key role of mG4-structure formation, rather than simple G-richness, in the aggregation process.

10) Would it be possible to map long range interactions in these constructs to strengthen that really these interactions are mediated by G-G base pairing? Alternatively can HiC or an alternative be done in ALS or FTD cells to monitor similar phenotypes as the pure in vitro system?

This is a very interesting point and indeed something we aim to establish on the long term. However, adapting HiC experiments to map mG4-structures require a substantial modification of the HiC protocol that we feel goes beyond the scope of this manuscript.

11) Other repeat expansion disease has strong phenotypes, not all of these repeats have G-rich repeats? Is this specific for ALS and FTD repeats, control repeats or additional repeats needs to be studied?

Again, a very interesting point raised by the reviewer. We already provided the control repeats suggested in **Figure 4**. Although this manuscript focuses on the characterisation of the $(GGGGCC)_n$ aggregation properties, we have preliminary evidence, generated with our collaboration with Prof Patani, suggesting that other repeat expansion can behave in a similar fashion. This is ongoing work that will be reported in a separate publication in due course.

12) The data presented in Figure 2 are clear but needs to be improved: the quality of the gel images needs to be greatly improved. Native PAGE should be utilized instead of agarose gels. A loading control should be included, or the ssDNA should be visualized to account for the presence of non-G4 bands. Noteworthy, increasing the salt concentration within a gel ("AGE") especially from 0 to 0.5 M already leads to a smile/shift of the samples; how do the authors account for this? Is the NMM and SYBR Safe (not "SYBRSAFE.") data from the same gel; this information the authors do not include. Also, the ladder in Fig. 1C is not labelled, which makes it hard to compare 1B and 1C. Also, a ladder should be included in 1D to make it possible to compare the images. Also, greater care should be taken in the methods section where it states 1.5 g agarose in 50 μ l TBE. I guess, the authors used a 50 ml mini-gel instead.

We apologise for the typos (SYBR Safe and 50 mL TBE instead of 50 μ L) in the methods section, which has now been amended accordingly. We have substantially improved the gel quality as requested, please refer to comments 3-5 of Reviewer 1. To address this, we used fluorescently labelled oligomers and a lower DNA annealing concentration to better visualise the distinct molecular bands. The new gel has now been added to the supplementary information of the manuscript (**Figure S1**). The ladder in NMM stained gels (**Figure 2D**) cannot be visualised as NMM is a specific dye for G4s, which does not fluoresce when exposed to a generic DNA ladder. As NMM, FAM and SYBR Safe present overlapping emissions and excitations, the imaging of different dyes was performed in separate gels, we have now amended the methods section to clarify this point.

13) In line 182 the statements “Having demonstrated the formation of multimolecular constructs” is a little bit of an over-interpretation, as the data in figure2 only show the presence of multiple different populations of folded oligodeoxynucleotides.

We agree with the reviewer that the data displayed in **Figure 2** alone is not sufficient to support this statement and is only with the combination of further data shown in the manuscript (NMM-photooxidation and analysis of control sequences) that this conclusion can be made. We have, therefore, reworded this statement accordingly.

Revised manuscript: “Having demonstrated the formation of multiple populations in samples of DNA (GGGGCC)_n, whose size increases with the concentration of K⁺ and the length of the repeat ...”

14) Which impact does crowding has on these events, So far the authors have used PEG to promote phase separation. Have they considered to monitor other methods (e.g. by pressure, crowing peptides, Ficoll, dextran) to induce crowing? Is the effect only due to crowing, as other molecules also have CD changes in crowing conditions and DNA dynamic change in crowing conditions, (e.g. Harve et al 2008, Mardoum et al 2018).

We agree with the reviewer that PEG is known to promote aggregation. For this reason, we tested the aggregation properties of (GGGGCC)₁₁ in absence of PEG to demonstrate this is a PEG-independent phenomena. As displayed in **Figure S6**, PEG aids but does not drive the aggregation on its own. This agrees with the fact that PEG favours the formation of multimeric G4s as previously reported in the literature. We have now added this evidence in the revised manuscript and attached here for convenience.

Figure S6: DNA (GGGGCC)₁₁ aggregates form in absence of the crowding agent PEG. Brightfield imaging of (GGGGCC)₁₁ annealed under mG4-forming conditions at 500 μ M. On the right, a faster cooling rate is used in presence of the PEG crowding agent. On the left, a slower cooling rate in absence of PEG. (See methods for details) 50 μ m scalebar.

15) Cloning and working with repeats of Guanine rich stretches is very difficult and the authors have established very good working methods for these repeats. However, it is essential to control the length of the repeats prior each experiment. I also recommend increasing the repeat size from n-24 to at least double or even higher to resemble the in vivo situation.

The sequences utilised for the studies were purchased from external providers and not cloned, which ensures quality control on the final length and sequence of the oligonucleotide. Unfortunately, DNA solid phase synthesis does not allow for the generation of longer sequences, especially considering the G-richness of these repeats, with $n = 12$ being the longest repeat length that can be purchased. However, the phase diagram displayed in **Figure 3D**, clearly reveal a trend by which the longer repeat lengths the lower is the concentration needed to observe aggregation, hinting that for repeat lengths $n > 24$ aggregation may be observed at biologically relevant concentrations, which is supported by the cellular data discussed in response 8.

16) Although the data are very interesting unfortunately the quality of Fig 3A is not sufficient to observe nucleic acid structure. Given the size bars in Fig. 3B and C the structure which is clearly visible in Fig. 3B and C should be approximately half as big in Fig. 1A. Perhaps a second panel, in which a stain was used similar to Fig. 3B,C would be helpful to actually visualize the structures in Fig. 3A.

We apologise, as there was a typo in the scalebar utilised in the images and the scale bar should have been 100 μ m, not 10 μ m. Nevertheless, we have now generated new confocal

images at higher magnification (20X) and added insets to ensure better visualisation of the structures as suggested in the revised **Figure 3A** (attached below for convenience).

Figure 3A: DNA (GGGGCC)_n aggregates in a G4-dependent fashion. Brightfield imaging of (GGGGCC)_n (n = 2-12) annealed under mG4-forming conditions at 250 μ M. 100 μ m scalebar.

17) Do the authors have any prove that NMM is not stabilizing, supporting these structures as NMM has been published as a G4 ligand?

NMM is added after the annealing process is complete and, therefore, the G4-stabilising effect of the dye does not affect the aggregate formation presented in the manuscript. This is further demonstrated in **Figure 3B**, where the aggregate is visualised by using FAM-labelled DNA and no NMM is present in solution.

18) In the PDS studied they showed that PDS can diminish mG4. As these data contrast with previous work and very interesting and strong. I recommend performing at least 2 additional G4 ligands in these assays. They suggest in the discussion the G4-G4 binding interaction is disturbed by the ligand. As they do not have proteins in their systems, it is tempting to speculate that the G4 itself is causing this effect and the ligand will prevent aggregation this opens a potential for new therapeutic applications. This speculation is very logic and elegant but based on the current data set too speculative and needs experimental prove. Have the authored considered that in the aggregates G-rich regions cluster together and do not form a G4?

Firstly, it is worth clarifying that our data is not in contrast with previous work. PDS has been shown to form more stable complexes with lower molecularity species over higher molecularity ones^{4,5}, suggesting that kinetics of formation could be affected in a similar way, which is why we have used this molecule to further validate this mechanism. We have tried using additional ligands, but their low solubility in aqueous buffer limits their use in our assay: because

experiments are carried out at high nucleic acid concentrations the required amount to ligands would be above the solubility threshold. To rule out aggregation mediated by G-rich regions rather than G4s, we provided controls sequences (i.e. GGGGTT) that have the same G-content but are less prone to form multimolecular G4s¹, demonstrating that no aggregation can be observed for these sequences. We confirmed that the aggregation was not hampered by the lack of G-C interaction by adding an additional control (GGGGAA) with the ability of forming mG4s but no G-C interactions (**Figure 4**). Furthermore, we have now demonstrated that aggregation can be promoted by addition of a G4-binding protein involved in ALS/FTD (TDP-43, see response to point 8), which allows us to robustly conclude that the aggregation mechanism described is G4-mediated.

19) Figure 4A, while panel 4C and b is of good quality and gel in D and E are not sufficient. there is no clear image visible for GGGTT and CCCC GG, also why do the scale bars differ in length. Here guidance in the text and improve figures are required to understand these data.

We have now generated new images for the panels displayed in **Figure 4E** to improve quality and have scalebars of the same size. The controls sequences GGGGTT and CCCC GG do not lead to aggregation, which also justify why no clear images are visible for these sequences.

Figure 4E: Role of base pairing and monomolecular G4s in the aggregation mechanism. Aggregation of the mutant sequences – Controls of $n = 6$ were annealed under mG4-forming conditions. Aggregates were only clearly distinguishable for (GGGGAA)₆ and (GGGGCC)₆ – the only sequences forming mG4s. 100 µm scalebar.

References

- Lago, S., Tosoni, E., Nadai, M., Palumbo, M. & Richter, S. N. The cellular protein nucleolin preferentially binds long-looped G-quadruplex nucleic acids. *Biochimica et Biophysica Acta (BBA) - General Subjects* **1861**, 1371–1381 (2017).
- Liano, D., Chowdhury, S. & Di Antonio, M. Cockayne Syndrome B Protein Selectively Resolves and Interact with Intermolecular DNA G-Quadruplex Structures. *Journal of the American Medical Society*. **143**, 20988–21002 (2021).
- Reddy, K., Zamiri, B., Stanley, S. Y. R., Macgregor, R. B. & Pearson, C. E. The Disease-associated r(GGGGCC)_n Repeat from the C9orf72 Gene Forms Tract Length-dependent Uni- and Multimolecular RNA G-quadruplex Structures*. *Journal of Biological Chemistry* **288**, 9860–9866 (2013).
- Bao, H.-L., Ishizuka, T., Iwanami, A., Oyoshi, T. & Xu, Y. A Simple and Sensitive 19F NMR Approach for Studying the Interaction of RNA G-Quadruplex with Ligand Molecule and Protein. *ChemistrySelect* **2**, 4170–4175 (2017).
- Santos, T., Salgado, G. F., Cabrita, E. J. & Cruz, C. G-Quadruplexes and Their Ligands: Biophysical Methods to Unravel G-Quadruplex/Ligand Interactions. *Pharmaceuticals (Basel)* **14**, 769 (2021).

REVIEWER COMMENTS

Reviewer #1 (Remarks to the Author):

The revised manuscript has highly improved. However, I have some concerns about the newly added experiments.

Figure 7: These data are very interesting, but they lack a negative control. Figure 7B shows that NMM enters the cells, but I would expect the same result for any other cell line. Did the authors perform the same experiment on a healthy cell line? The legend to Figure S12 states "Cells fluorescing in the NMM fluorescence channel were counted in the C9 cell line and healthy neuronal cell controls", but there are no data for healthy neuronal cell controls. Please add controls.

I disagree with the sentence "The occurrence of G4s in neurons can be also captured by flow cytometry (Figure S12), which showed a notably enhanced fluorescent intensity upon NMM (10 μ M) treatment (Figure 8B)." The flow cytometry experiment only proves that NMM can efficiently enter in the neurons, as stated in the description of Figure 7B "NMM treatment resulted in detection of fluorescence of the characteristic NMM spectrum, indicating efficient incorporation into motor neurons". Furthermore does "G4s" in this sentence mean all G4s or just mG4s?

In Figure 7A the staining with NMM is puzzling: I would expect the signal to be only in the cytoplasm. I think that repeating the staining on healthy neurons will confirm the formation of mG4 once and for all. The DMSO control is correct, but it alone cannot be considered as a negative control for the presence of mG4 in C9 cells.

I disagree with the sentence "NMM staining has revealed the presence of G4-structures in pathological aggregates, suggesting that the model described in this study could be of both pathophysiological and therapeutic relevance." How can you be sure that NMM stains pathological aggregates (and only them) as no co-staining with pathological proteins was performed?

A minor concern about the Results of figure 6C: does TDP-43 RRM1-2 forms itself aggregates? A clearer sentence would help the reader.

Reviewer #2 (Remarks to the Author):

The authors have addressed all my comments and concerns and I would like to congratulate them on this nice study.

Point by point response to the reviewers.

Reviewer #1

1) Figure 7: These data are very interesting, but they lack a negative control. Figure 7B shows that NMM enters the cells, but I would expect the same result for any other cell line. Did the authors perform the same experiment on a healthy cell line? The legend to Figure S12 states "Cells fluorescing in the NMM fluorescence channel were counted in the C9 cell line and healthy neuronal cell controls", but there are no data for healthy neuronal cell controls. Please add controls.

We agree with the reviewer that this is a critical control to strengthen to our conclusions. We now provide data directly comparing *C9orf72* mutant neurons with healthy control motor neurons. Importantly, we reveal a statistically significant increase in the number of NMM foci both in the cell soma and within the nucleus in *C9orf72* mutant motor neurons when compared to control motor neurons. Notably, no differences in the intensity of NMM signal, the intensity and size of NMM foci, and the prevalence of cells with NMM foci were detected, suggesting that differences in foci numbers cannot be sufficiently explained by a systematic bias. Given that G4s are generally abundant in cells (e.g., at the telomeres and promoters), NMM signal in control cells is to be expected, which is consistent with previous literature in this area (Biffi *et al. Nat. Chem.* 2013; Henderson *et al. Nucleic Acis Res.* 2017; Di Antonio *et al Nat. Chem.* 2020). We have now added these data in **Figure 7** of the revised manuscript and in **Figure S12**, which we have added below for convenience.

Figure 7: *C9orf72* mutant iPSC derived motor neurons exhibit increased number of G4 foci per cell soma and increased percentage of G4 foci within the nucleus, when compared to healthy controls. A. Confocal representative images of iPSC derived motor neurons labelled with Nuclear Stain DRAQ5 and NMM. Upon staining with NMM, both control and *C9orf72* mutant cells present fluorescent G4 foci. B. Quantification of the

number of NMM foci per cell soma in control and mutant motor neurons. *C9orf72* mutant cells exhibit a significant increase in the number of NMM foci per cell, *p*-value from Mann-Whitney U test. **C. Quantification of the percentage of NMM foci in the nucleus in control and mutant motor neurons.** Mutant cells exhibit a significant increase in the percentage of NMM foci within the nucleus, *p*-value from unpaired two-tailed t-test, mean \pm s.d. **p* < 0.05, ***p* < 0.01.

Figure S12. NMM is efficiently incorporated into iPSC derived motor neurons. **A.** Confocal representative images of iPSC derived motor neurons stained with neuronal marker β -Tubulin III (green) and nuclear dye DRAQ5 (red). **B.** Confocal representative images of motor neurons treated with DMSO and labelled with nuclear stain DRAQ5 (red). DMSO-treated control and *C9orf72* mutant motor neurons exhibit no autofluorescence in the NMM spectrum. **C.** Quantification of the NMM signal intensity in DMSO- and NMM-treated control and *C9orf72* cells. NMM is efficiently incorporated into iPSC derived motor neurons and specifically detected within its emission spectrum, *p*-value from two-way ANOVA with Tukey correction for multiple comparisons. **D.** Quantification of the percentage of neurons containing NMM foci in DMSO- and NMM-treated control and *C9orf72* cells. Cells treated with NMM exhibit positive NMM foci across the whole population, *p*-value from two-way ANOVA with Tukey correction for multiple comparisons. **E.** Comparison of NMM foci intensity between NMM-treated control and *C9orf72* cells. No difference in NMM intensity is observed between the two populations, *p*-value from unpaired two-tailed t-test. **F.** Comparison of NMM foci size between NMM-treated control and

C9orf72 cells. No difference in NMM foci size is observed between the two populations, p -value from Mann-Whitney U test, mean \pm s.d. **** $p < 0.0001$.

We have also added the flow cytometry data for both control and mutant cells, which we are inserting here for convenience:

A.

B.

Figure S13. Flow cytometry data for iPSC-derived motor neurons treated with DMSO and NMM. A. Representative flow cytometry density plots for DMSO and NMM treatments in control and C9orf72 mutant cells. The plots indicate efficient incorporation of NMM into treated cells and minimal autofluorescence; **B. DMSO and NMM treatment fluorescence histograms**, where control (Left) and mutant (Middle) samples are compared individually, demonstrating that NMM was incorporated into the cells and the fluorescence is recorded in the characteristic NMM spectrum; Merged graph (Right) of DMSO and NMM fluorescence histograms in control and mutant cells. The intensity recorded in the mutant cells does not differ to that of their control counterparts.

2) I disagree with the sentence “The occurrence of G4s in neurons can be also captured by flow cytometry (Figure S12), which showed a notably enhanced fluorescent intensity upon NMM (10 μ M) treatment (Figure 8B).” The flow cytometry experiment only proves that NMM can efficiently enter in the neurons, as stated in the description of Figure 7B “NMM treatment resulted in detection of fluorescence of the characteristic NMM spectrum, indicating efficient incorporation into motor neurons”. Furthermore does “G4s” in this sentence mean all G4s or just mG4s?

We agree with the reviewer and indeed, flow cytometry was employed to ensure that NMM has been efficiently incorporated into the cells and that no auto-fluorescence could be detected in the NMM-emission range. We have now clarified this in the text and the figure legend (Figure S13) and the sentence mentioned by the reviewer has been removed accordingly. Given that NMM cannot discriminate between unimolecular or multimolecular G4s, we cannot ascribe its staining to mG4s specifically, which is why we use the generic term G4-staining.

3) In Figure 7A the staining with NMM is puzzling: I would expect the signal to be only in the cytoplasm. I think that repeating the staining on healthy neurons will confirm the formation of mG4 once and for all. The DMSO control is correct, but it alone cannot be considered as a negative control for the presence of mG4 in C9 cells.

As mentioned in our answers to point 1 and 2, NMM cannot selectively stain mG4s and endogenous G4s are also highly abundant in the nucleus, as extensively reported in the literature (Biffi *et al. Nat. Chem.* 2013; Henderson *et al. Nucleic Acids Res.* 2017; Di Antonio *et al. Nat. Chem.* 2020), so it would be expected to observe NMM staining in the nucleus. Moreover, RNA *foci* from expansion repeats are known to be detected in the nucleus (Simone *et al. EMBO Mol. Med.* 2018), which is in line with our observations.

4) I disagree with the sentence “NMM staining has revealed the presence of G4-structures in pathological aggregates, suggesting that the model described in this study could be of both pathophysiological and therapeutic relevance.” How can you be sure that NMM stains pathological aggregates (and only them) as no co-staining with pathological proteins was performed?

We agree with the reviewer that this was an over-optimistic interpretation of our data. To address this concern, we have now rephrased as follows: ‘NMM staining revealed higher prevalence of G4-structures in *C9orf72* mutant motor neurons compared to control counterparts, suggesting that the model proposed in this study could be of both pathophysiological and therapeutic relevance.’

5) A minor concern about the Results of figure 6C: does TDP-43 RRM1-2 forms itself aggregates? A clearer sentence would help the reader.

TDP-43 RRM1-2 is known to aggregate on its own right, forming nanoparticles that cannot be detected *via* confocal microscopy. In our experiment, we showed that the macroscopic aggregates that are detected *via* confocal microscopy can only be observed when the protein

is exposed to the nucleic acid expansion repeat. This is why to ensure that TDP-43 RRM1-2 was behaving as previously reported in the literature we also performed TEM to assess nanoscale aggregates as displayed in **Figure S11**.

The text has been reworded to allow for a clearer understanding as follows:

“Importantly, no macroscopic aggregates can be detected with TDP-43 RRM1–2 alone. This suggests that the emergence of larger aggregates requires both the protein and the nucleic acid components. The formation of TDP-43 RRM1–2 aggregates that cannot be detected by confocal microscopy was assessed *via* transmission electron microscopy, revealing the formation of microscopic aggregates that cannot be detected by confocal microscopy, consistently with previous literature⁵⁶ (**Figure S11**).”